# Quality by Design-Based Development of Solid Self-Emulsifying Drug Delivery System (SEDDS) as a Potential Carrier for Oral Delivery of Lysozyme

**DOI:** 10.3390/pharmaceutics15030995

**Published:** 2023-03-20

**Authors:** Merima Šahinović, Alharith Hassan, Katalin Kristó, Géza Regdon, Edina Vranić, Tamás Sovány

**Affiliations:** 1Department of Pharmaceutical Technology, Faculty of Pharmacy, University of Sarajevo, Zmaja od Bosne 8, 71000 Sarajevo, Bosnia and Herzegovina; 2Institute of Pharmaceutical Technology and Regulatory Affairs, University of Szeged, Eötvös u 6., 6720 Szeged, Hungary

**Keywords:** quality by design, self-emulsifying drug delivery systems, SEDDS, liquid SEDDS, solid SEDDS, hydrophobic ion pairing, oral delivery of biopharmaceuticals

## Abstract

For many years, researchers have been making efforts to find a manufacturing technique, as well as a drug delivery system, that will allow for oral delivery of biopharmaceuticals to their target site of action without impairing their biological activity. Due to the positive in vivo outcomes of this formulation strategy, self-emulsifying drug delivery systems (SEDDSs) have been intensively studied in the last few years as a way of overcoming the different challenges associated with the oral delivery of macromolecules. The purpose of the present study was to examine the possibility of developing solid SEDDSs as potential carriers for the oral delivery of lysozyme (LYS) using the Quality by Design (QbD) concept. LYS was successfully ion paired with anionic surfactant, sodium dodecyl sulphate (SDS), and this complex was incorporated into a previously developed and optimized liquid SEDDS formulation comprising medium-chain triglycerides, polysorbate 80, and PEG 400. The final formulation of a liquid SEDDS carrying the LYS:SDS complex showed satisfactory in vitro characteristics as well as self-emulsifying properties (droplet size: 13.02 nm, PDI: 0.245, and zeta potential: −4.85 mV). The obtained nanoemulsions were robust to dilution in the different media and highly stable after 7 days, with a minor increase in droplet size (13.84 nm) and constant negative zeta potential (−0.49 mV). An optimized liquid SEDDS loaded with the LYS:SDS complex was further solidified into powders by adsorption onto a chosen solid carrier, followed by direct compression into self-emulsifying tablets. Solid SEDDS formulations also exhibited acceptable in vitro characteristics, while LYS preserved its therapeutic activity in all phases of the development process. On the basis of the results gathered, loading the hydrophobic ion pairs of therapeutic proteins and peptides to solid SEDDS may serve as a potential method for delivering biopharmaceuticals orally.

## 1. Introduction

Biopharmaceuticals have been effectively employed in the treatment of many diseases (malignant tumors, diabetes, and cardiovascular and autoimmune diseases) due to their high specificity and safety. The development of drug delivery systems for biopharmaceuticals faces several obstacles and risks due to their complexity and heterogeneous nature. For many years, researchers have struggled to find a manufacturing technique, as well as a drug delivery system, that will allow for oral delivery of proteins and peptides to their target site of action without affecting their biological activity [1,2,3].

Lipid-based nanocarriers appear to be of particular relevance when it comes to developing oral biopharmaceutical formulations, which have been the subject of many studies so far. These delivery systems have the potential to overcome all main physiological barriers to the oral delivery of biopharmaceuticals, including the enzymatic, sulfhydryl, mucus, and epithelial barrier [4].

Among lipid-based formulations, self-emulsifying drug delivery systems (SEDDSs) show encouraging in vivo results and great potential as carriers for oral delivery of biopharmaceuticals. These systems consist of lipids, surfactants, and cosolvents, typically firstly referred to as liquid preconcentrates. Gentle agitation of liquid preconcentrates provided by the digestive motility of the stomach and intestine is sufficient to emulsify these formulations upon oral administration and form oil/in water micro- or nanoemulsions [5].

SEDDSs can be readily manufactured by simply mixing components, making them ideal for industrial applications because large batches may be generated without the use of complicated technologies.

The manufacturing process itself does not contain any high-energy operations, which can be of great interest for biopharmaceuticals [6]. Additionally, to address issues regarding the physical and chemical stability of liquid SEDDSs, different solidification techniques can be used to transform liquid SEDDSs into free-flowing powders, which can be further processed into other solid dosage forms, such as tablets or pellets [7,8,9].

Recent research has demonstrated that SEDDSs can be effective carriers for the oral administration of hydrophilic peptides and proteins by shielding them from the GIT environment and enhancing their penetration across the intestinal mucus barrier by reducing peptide–mucus interactions [10]. If the biopharmaceuticals are loaded inside the oil droplets of SEDDSs, they can be effectively protected towards proteolytic activity and/or can act as permeation enhancers, leading to improved bioavailability [11,12].

However, due to their hydrophilicity, their incorporation into lipid systems can be challenging, thus increasing their lipophilicity using methods, such as the formation of hydrophobic ion pairs (HIPs), chemical modification, or the double emulsification process, which must be considered as the first step in formulation development [13]. Hydrophobic ion pairing (HIP) is the most common technique and uses non-covalent electrostatic interaction between an ionized hydrophilic drug and an oppositely charged hydrophobic counterion to disguise undesirable physicochemical characteristics [14,15]. The HIP approach possesses numerous advantages: it is simple, inexpensive, it does not involve chemical modification of the drug, and biological efficacy and drug safety are preserved [13].

Incorporating HIPs in SEDDSs appears to be a potential strategy for significantly increasing the oral bioavailability of hydrophilic drugs, such as matrine [16] and morine [17], and charged hydrophilic drugs, such as enoxaparin [18], atazanavir [19], vancomycin [20], cromolyn sodium [21], polymyxin B [22], etc. More crucially, promising in vivo results of SEDDSs containing ion-paired biopharmaceuticals were obtained [23,24,25]. However, no lipid formulation of this type has reached the stage of clinical trials, which emphasizes the need for further intensive research in this field.

One of the first studies using this concept was conducted by Zhang et al., where insulin was complexed with soybean phosphatidylcholine (SPC) and incorporated into an SEDDS formulation, after which oral administration to diabetic rats generated a favorable hypoglycemic effect and increased relative bioavailability of insulin up to 7.15%, compared to 0.11% of the insulin solution [26]. In their work, Hintzen et al. reported using leuprolide as a model peptide drug and sodium oleate as the counterion. After incorporating the leuprolide:oleate complex in microemulsion droplets (SMEDDs), an even 17.2-fold increase in oral bioavailability in rats compared to an aqueous leuprolide solution was achieved. The formulation was able to protect leuprolide oleate from enzymatic degradation by trypsin and α-chymotrypsin [27].

Bonengel et al. reported, in their in vivo study on pigs, that an SEDDS loaded with octreotide-deoxycholate and octreotide-docusate exhibited a permeation-enhancing effect as well as 17.9-fold and 4.2-fold higher bioavailability versus control. Additionally, they also concluded that oral uptake of peptides from an SEDDS into the systemic circulation is strongly influenced by the type of HIP, as their findings suggest that there was no improvement in oral bioavailability in the case of the octreotide-decanoate complex, which could be attributed to the different release behavior and intrinsic properties of the anionic surfactants used [23].

Menzel et al. developed an oral SEDDS of exenatide, a potent glucagon-like peptide-1 (GLP-1) analogue, following hydrophobic ion paring with sodium docusate (DOC). Orally administered exenatide/DOC SEDDS showed a relative bioavailability (versus subcutaneous injection) of 14.62% and caused a significant decrease in AUC values of blood glucose levels in rats [28]. Relative oral bioavailabilities of 19.6% and 15.2% (versus subcutaneous injection) were obtained for orally administered exenatide-n-octadecyl sulfate (SOS)-loaded SEDDS and exenatide-DOC-loaded SEDDS, respectively, demonstrating the importance of counterion type for oral peptide administration using the HIP strategy [24].

The potential of HIP-loaded SEDDS as a promising carrier for oral vaccination has been shown in research by Lupo et al. Model antigen bovine serum albumin (BSA) was successfully incorporated in an SEDDS after ion pairing with different counterions and administered orally to mice. Formulations induced a strong systemic (anti-BSA-IgG titer) and mucosal (anti-BSA-IgA titer) immune response after oral administration, suggesting that this might be a game-changing technique for oral vaccination [29].

For more detailed information on recent progress in hydrophobic ion pairing and lipid-based drug delivery systems for oral delivery of biopharmaceuticals, the reader is referred to the following comprehensive review articles: [30,31].

Despite the foregoing, the full potential of an SEDDS including HIPs for oral biopharmaceutical delivery should yet be achieved, since some crucial aspects still have to be addressed. It was, therefore, the aim of this study to use the Quality by Design (QbD) approach to develop, optimize, and evaluate solid SEDDSs as carriers for the oral delivery of lysozyme (LYS) as a model drug. From its discovery by A. Fleming in 1921, LYS (or muramidase or N-acetylmuramic acid hydrolase E.C. 3.2.1.17) has attracted attention as an effective antimicrobial agent. Through the hydrolysis of the 1,4-glycosidic bonds between N-acetylglucosamide (NAG) and N-acetylmuramic acid (NAM) in the polysaccharide backbone of the peptidoglycans of the Gram-positive bacterial cell wall, LYS exerts its antimicrobial activity [32]. This basic protein is composed of 129 amino acids and is naturally presented in the human body (saliva, tears, mucus) as well as tissues of animals and plants.

LYS is available on the market as a conventional tablet and syrup; however, there are limited data available on its pharmacokinetics in humans. Studies have suggested that it is absorbed from the intestine in humans, although the extent of absorption is not as high [33,34,35]. It was found that LYS absorption in the gut was segment-specific and it is absorbed preferentially from the upper part of the intestine, most likely through clathrin-mediated endocytosis [33]. Therefore, its successful formulation in a stable oral solid dosage form with adequate bioavailability may contribute to managing many diseases.

The QbD concept is a knowledge- and risk-assessment-based quality management approach, used in pharmaceutical research and development. The QbD, or the “GMP of the 21st century”, is a methodology where the product and the manufacturing process are designed and developed according to the previously defined expectations [36]. This holistic and systemic approach will be used in all parts of this work, in order to identify all risks as well as critical material attributes and process parameters whose variability has a critical effect on the aimed product quality.

## 2. Materials and Methods

### 2.1. Materials

Lyophilized LYS from chicken egg white (Mw: 14.3 kDa) (MedChemExpress, Monmouth, NJ, USA) was used as a model protein. Lyophilized Micrococcus lysodeikticus (Sigma-Aldrich, St. Louis, MO, USA) was used as Gram-positive bacteria to investigate LYS enzymatic activity. Sodium dodecyl sulphate (SDS) (Molar Chemicals Ltd., Budapest, Hungary) was used as counterion for HIP and sodium hydroxide and hydrochloric acid (Ph. Eur.) as pH adjusters. Tween 80^®^ (polysorbate 80) (Merk KgaA, Darmstadt, Germany), PEG 400 (polyethylene glycol 400) (Molar Chemicals Ltd., Budapest, Hungary), and Miglyol 812^®^ (medium-chain triglycerides) (Sasol Germany GmbH, Witten, Germany) were used in preparation of liquid SEDDS. Neusilin^®^ UFL2 (magnesium aluminometasilicate) (Fuji Chemical Industries Co., Ltd. Kamiichi, Nakaniikawa, Toyama, Japan), Syloid^®^ 244 FP, and Syloid^®^ AL-1 FP (Grace Davison, Worms, Germany) were used as solid carriers in preparation of solid SEDDS. Vivapur 102^®^ (microcrystalline cellulose) and Vivasol^®^ (croscarmellose sodium) (JRS PHARMA GmbH & Co. KG, Rosenberg, Germany) were used in tablet formulation as diluent and disintegration agents, respectively. Salts used to prepare the phosphate buffer as well as all the other reagents were of analytical grade.

### 2.2. Methods

#### 2.2.1. Quality by Design Methodology

According to the guidelines of the International Council of Harmonisation of Technical Requirements for Pharmaceuticals for Human Use [36,37,38], the QbD methodology comprises the following steps: defining the Quality Target Product Profile (QTPP), identification of Critical Quality Attributes (CQAs), identification of Critical Material Attributes (CMAs), and Critical Process Parameters (CPPs), performing initial risk assessment (RA), application of a credible and robust statistical Design of Experiment (DoE), the establishment of the Design Space (DS), and definition of the control strategy.

The initial RA was performed to evaluate the risks and obtain the Risk Priority Number (RPN). The severity, probability, and vulnerability of the various parameters were combined to compute the overall risk. Then, the interdependence rating amongst the CQAs and the CMAs/CPPs was performed using a three-level scale to describe the relationship between parameters. The relations were labelled as low, medium, or high. Additionally, during knowledge space development, to investigate the cause-and-effect relationships, Ishikawa fish bone diagrams were set up for all steps of the experimental work.

After conducting a DOE analysis, statistical equations are often generated to summarize the results. The equations (y = b_0_ + b_1_x_1_ + b_2_x_2_) can be used to predict the outcome (y) for any combination of input variable values (x_1_, x_2_). The coefficients of the factors describe the change in the CQA if the factor value is increased from the 0 to the +1 level, with calculation on the basis of linear regression. The alpha value indicating significant factors was 0.05 and the confidence interval was 95%. If the influence was statistically significant, the factor in the equation is highlighted in bold (*p* < 0.05).

#### 2.2.2. Development and Characterization of HIP Complex—LYS:SDS

##### Hydrophobic Ion Pairing of LYS with SDS

The method was performed similarly to that described previously [39,40]. Briefly, 80 mg of LYS was completely dissolved in 20 mL of phosphate buffer solution (pH 6.8) to obtain a final concentration of 4 mg/mL, and the pH of the LYS solution was adjusted with 0.1 M HCl or NaOH according to the experimental design matrix (Table 1). The chosen counterion, sodium dodecyl sulphate (SDS), was dissolved in the same media, in concentrations of 0.161 mg/mL, 0.322 mg/mL, and 0.484 mg/mL for the preparation of 1:2, 1:4, and 1:6 molar ratio complexes, respectively. Thus, 1 mL of the given SDS solution was added in a drop-wise manner to 1 mL LYS solution to obtain the different molar ratios according to the experimental design (Table 1). The spontaneous development of a cloudy suspension was a visual indication of complex formation. The precipitated LYS:SDS complexes were isolated by centrifugation (10 min, 8000 rpm) using Hermle Z323K high-performance refrigerated centrifuge (Hermle AG, Gossheim, Germany). The resulting HIP complexes were then freeze-dried (Scanvac Coolsafe laboratory freeze-dryer (LaboGene, Lillerød, Denmark)) at −20 °C for 24 h under a pressure of 1.3 Pa, and then kept at 25 °C for 24 h for secondary drying to obtain lyophilized powders. Uncomplexed LYS was determined by analyzing the amount of remaining LYS in the supernatant using UV/VIS spectrophotometry (Spectronic Helios Alpha, 084304, Spectronic Unicam, Cambridge, UK). The samples were stored at −10 ± 2 °C until further investigations.

##### Experimental Design—Full Factorial Design (23)

Two-level five-factor fractional factorial design (2^5−2^), with a total of 8 experiments, was used to screen the effect of five independent variables (molar ratio, pH, mixing speed, mixing time, and temperature) on complexation efficacy (Appendix A). According to the results of the screening design, three of the five independent variables, namely molar ratio (x_1_), pH (x_2_), and temperature (x_3_), were further investigated using a full factorial 2^3^ design, with a total of 8 experiments and 1 central point (Table 1). Binding efficiency (y_1_) and enzyme activity (y_2_) were selected as optimization parameter responses. Analysis was performed using Tibco Statistica v.13.5. (Tibco Statistica Inc, Palo Alto, CA, USA) software.

##### Determination of LYS Binding Efficiency

In order to calculate LYS binding efficiency, uncomplexed LYS remaining in the supernatant after centrifugation was measured using a UV spectrometer (Spectronic Helios Alpha, 084304, Spectronic Unicam, Cambridge, UK) at lambda max. 280 nm for each sample based on a pre-recorded calibration line.

The percentage of binding efficiency (% BE) of LYS was calculated by the following equation (Equation (1)):(1)$$\%\;\mathrm{BE}=100-\frac{\left(C_{afterHIP}\right)}{\left(C_{beforeHIP}\right)}\times 100$$
where *C_before_ HIP* and *C_after_ HIP* are the initial concentration of the enzyme and the amount of unreacted enzyme remaining in the supernatant, respectively.

##### Dissociation of LYS from HIP Complex

Then, 2 mg of the prepared lyophilized LYS:SDS complexes (C1–C9) was dissolved in 1 mL PBS (pH 6.8) containing 137 mM NaCl and incubated for 4 h, 24 h, or 168 h at room temperature to study complex dissociation. After centrifugation, the amount of LYS dissociated from the complex was measured using a UV/VIS spectrophotometer (Spectronic Helios Alpha, 084304, Spectronic Unicam, Cambridge, UK), and dissociation percentage was calculated according to the following equation (Equation (2)):(2)$$Dissociation\left(\%\right)=\left(\frac{amount\;of\;LYS\;in\;supernatant}{initial\;amount\;of\;LYS\;in\;HIPs}\right)\times 100$$

##### Evaluation of LYS Enzymatic Activity after HIP

Firstly, LYS:SDS complexes (C1–C9) were incubated with 1 mL PBS (6.8 pH) for 24 h at room temperature to force complex dissociation. After centrifugation (8000 rpm, 10 min), the enzymatic activity of LYS was measured in the supernatant to ensure that the complexation process did not negatively affect enzyme activity. The activity of LYS was determined by measuring the degradation of lyophilized micrococcus lysodeikticus using a UV spectrometer (Genesys 10 S UV-VIS Spectrometer, Thermo Fisher Scientific Inc., Waltham, MA, USA). Briefly, 0.25 mg/mL bacterial suspension of M. lysodeikticus (Sigma-Aldrich, St. Louis, MO, USA) was prepared in 100 mL phosphate buffer (pH 6.8). The basic absorbance of the suspension at 450 nm was ∼0.7. Then, 100 µL of dissociated LYS solution was added to 2.5 mL of bacterial suspension and mixed with inversions for 20 s in a quartz cuvette. Since digestion of bacterial membrane causes a decrease in the absorbance at 450 nm, the change in bacterial absorption was measured every 5 s during a total incubation period of 5 min at 25 °C. The activity of the samples was compared to a freshly prepared LYS solution in the same concentration range. The fitted exponential decay curves were used to calculate the percentage of enzyme activity.

##### Fourier-Transform Infrared Spectroscopy (FTIR)

To characterize the complex formation and secondary structure of LYS in the HIP complexes, the infrared spectra of the LYS, SDS, their physical mixtures, and freeze-dried LYS:SDS complexes were obtained using an FT-IR (Avatar 330 FT-IR, Thermo Fisher Scientific Inc., Waltham, MA, USA ) spectrophotometer with potassium bromide pressed disc method. Samples were scanned for absorbance in a wavelength range of 600–4000 cm^−1^. Collected spectra represent the average of 128 individual scans with a spectral resolution of 4 cm^−1^ and applying CO_2_ and H_2_O corrections. The SpectraGryph (version 1.2.15.; Dr. Friedrich Menges Software, Entwicklung, Germany) was used in evaluation of the results.

#### 2.2.3. Development and Characterization of Liquid SEDDS Loaded with LYS:SDS Complex

##### Screening of Oils, Surfactants, and Co-Surfactants

Miglyol 812^®^ (medium-chain triglycerides), Tween 80^®^ (polysorbate 80), and PEG 400 (polyethylene glycol 400) were chosen as suitable oil, surfactant, and co-surfactant for SEDDS development. Selected excipients are often utilized in SEDDS formulations and have been regarded as safe (GRAS status). Preliminary screening of the solubility of optimized LYS:SDS complex in each component was performed, namely by adding 0.5, 1, 1.5, 2, 2.5, or 3 mg complex to 1 mL of each excipient and stirred at 37 °C for 2 h to achieve complete dissolution of the complex. When a clear oily solution was formed, the complex was assumed to be completely soluble in the excipient. The best solubility of the complex was found in Tween 80 (3 mg was dissolved), followed by PEG 400 (max. 2 mg) and Miglyol (max. 1.5 mg).

##### Preparation of Liquid SEDDS According to the Mixture Design

A constrained three-factor mixture experimental design and response surface methodology was implemented to investigate the effects of selected excipients and their interactions on SEDDS properties and optimize the incorporation of the complex in liquid SEDDS. Percentage of oil phase (A), the percentage of surfactant (B), and the percentage of co-surfactant (C) were used as independent variables with a total concentration of 100%. The droplet size (y_3_), polydispersity index (y_4_), and zeta potential (y_5_) were chosen as responses. Respective low and high levels of each independent variable were selected based on previously reported (pseudo)ternary diagrams, as well as usually used ratios of components in SEDDS development. Thus, 5–40% oil, 40–75% surfactant, and 20–50% co-surfactant were used as upper and lower limits [41,42]. Nine experiments were designed according to a constrained ternary mixture design by Tibco Statistica v.13.5. (Statsoft, Tulsa, OK, USA) software (Table 2). The oil phase and surfactants were each mixed in different weight ratios (Smix ratios (a total of 9 formulations)) according to mixture design matrix, as shown in Table 2. Components were mixed with a magnetic stirrer (700 rpm) at 45 °C for 2 h to obtain homogenous mixtures.

##### Evaluation of Self-Emulsifying Properties

Prepared liquid preconcentrates were emulsified and their self-emulsifying ability was investigated. Thus, 0.1 mL of prepared blank liquid preconcentrates was dispersed with distilled water (1:100) by gentle agitation (150 rpm) using a magnetic stirrer at 37 °C. These mixing conditions were chosen to mimic the in vivo conditions. All samples were visually observed for clarity, phase separation, and coalescence of oil droplets, immediately after preparation as well as after storing for 7 days at room temperature. The grading system proposed by Singh et al. was used to define the appearance of resulting emulsions after visual observation [43]. Briefly, rapidly forming emulsion (less than 1 min), which is transparent or slightly bluish, was noted as Grade I emulsion, while the less-clear and bluish emulsion was referred to as Grade II. Dark-bluish to milky emulsion forming within 2 min was defined as Grade III, while Grade IV was characterized by slowly forming (longer than 3 min) and greyish-white appearance. Formulation with large oil droplets on the surface with poor or minimal emulsification property refers to Grade V emulsion.

The transmittance (%) of the formed emulsions was determined using a UV spectrophotometer (Genesys 10 S UV-Vis Spectrometer, Thermo Fisher Scientific Inc., Waltham, MA, USA) at 638 nm against distilled water as the blank.

##### Droplet Size and Zeta Potential Measurements

The mean droplet size, polydispersity index (PDI), and zeta potential of the resulting emulsions were determined by photon correlation spectroscopy (PCS), using Malvern Nano ZS Zetasizer (Malvern Instruments Ltd., Malvern, UK) equipped with He-Ne laser (633 nm). Measurements were performed in disposable folded capillary cuvettes (Malvern Instruments, UK) at 25 °C, with an equilibration time of 120 s and back-scattering detection at 173°, in the general data processing model. Refractive index (RI) was determined before the measurement (RI: 1334).

##### Stability Studies of SEDDS

The kinetic stability of SEDDS formulations was determined by performing a centrifugation test. After emulsification in distilled water (1:100), formulations were centrifuged (5 min at 10,000 rpm) to visually examine the samples for phase separation or turbidity, creaming, or cracking, which would indicate instability in the preparation. In addition to visual observation, formulation stability was also assessed by measuring droplet size and zeta potential, after a storage time of 7 days at 25 °C. The samples were also subjected to different stress conditions, such as freeze–thaw cycle (−20 °C and +25 °C) and heating–cooling cycle (4 °C and 45 °C), which were all followed by visual inspection.

##### Preparation of LYS:SDS-Complex-Loaded SEDDS Formulations

Increasing concentrations of optimized LYS:SDS complex, namely 0.5%, 1%, 2%, 3%, and 5% (*m*/*v*%) were added to the previously homogenized liquid preconcentrates and stirred at 50 °C for 2 h to achieve complete dissolution of the complex. When a clear oily solution was formed, the complex was assumed to be completely soluble in the liquid preconcentrate. Samples were centrifuged (5 min, 10,000 rpm), and observed for any visual signs of precipitation or phase separation. SEDDSs were stored at room temperature, protected from the light, for further investigations. Maximum payload (%, *w*/*w*) as well as corresponding LYS payload (%) were calculated according to the following equations (Equations (3) and (4)) [44]:(3)$$payload\;complex\left(\%\right)=\frac{m_{dissolved\;complex}}{m_{SEDDS\;preconcentrate}}\times 100$$
(4)$$payload\;LYS\left(\%\right)=\frac{m_{LYS}}{m_{LYS}+m_{SDS}}\times payload\;complex$$

##### Robustness to Dilution and pH Changes

It was reported that different dilution ratios, as well as different pH values of the diluent, may affect the physical stability of SEDDSs. To evaluate robustness to dilution as well as stability of liquid SEDDSs under simulated in vivo conditions (under gastric and intestinal conditions), optimized liquid preconcentrate was diluted in 1:100 and 1:1000 ratio in distilled water (DW), PBS pH 6,8 and 0.1 M HCl pH 1,4. All diluents were filtered before the test. Samples were incubated at 37 °C with slight agitation (150 rpm), and their droplet size and polydispersity index (PDI) were determined as described above.

##### Determination of log *D_SEDDS/release medium_*

Partition coefficient (log *D_SEDDS/release medium_*) was determined by measuring the solubility of the LYS:SDS complex in SEDDS preconcentrate and different release mediums. An increasing amount of complex was dissolved in respective SEDDS preconcentrate, and the maximum payload of the complex was considered as its solubility in SEDDS (*C_SEDDS_*). Solubility of the complex in release mediums (*C_RM_*) was determined by dissolving lyophilized LYS:SDS complex in 1 mL of distilled water, phosphate buffer pH 6.8, or 0.1 M HCl by stirring at 300 rpm at 37 °C for 3 h. Following this, the samples were centrifuged (10,000 rpm, 5 min) (Hermle Z323 K high-performance refrigerated centrifuge (Hermle AG, Gossheim, Germany)), and the amount of dissolved complex in the supernatant was analyzed using UV/VIS spectrophotometry (Spectronic Helios Alpha, Spectronic Unicam, Cambridge, UK) [45].

Log $D_{\frac{SEDDS}{RM}}$ of the complex was calculated according to Equation (5):(5)$$\log D_{\frac{SEDDS}{RM}}=\frac{\log C_{SEDDS}}{\log C_{RM}}$$

The percentage of complex remaining in SEDDS droplets (*C_oil droplets_*), as well as the percentage of complex release into the release medium (*C_release medium_*), can be determined using Equations (6) and (7):(6)$$C_{oil\;droplets}\left(\%\right)=\left(\frac{100}{1+\frac{V_{release\;medium}}{V_{SEEDS}\;*\;D_{SEEDS/RM}}}\right)$$
(7)$$C_{release\;medium}\left(\%\right)=100-C_{oil\;droplets}$$
where the *V_release medium_* is the volume of the release medium, *V_SEDDS_* is the volume of the SEDDS preconcentrate, and *D_SEDDS_*_/*release medium*_ represents the distribution coefficient of the complex in SEDDS preconcentrate and release medium [45].

#### 2.2.4. Development and Characterization of Solid SEDDS Loaded with LYS:SDS Complex

##### Preparation of Solid SEDDS Containing LYS:SDS Complex

Optimized liquid SEDDS was transformed into solid powder by adsorbing onto the different hydrophobic solid carriers, Neusilin^®^ UFL2, Syloid^®^ 244FP, and Syloid^®^ AL-1 FP. Them, 1 g of liquid SEDDS was placed in a mortar with a smooth surface, and increasing amounts of the solid carrier were added and mixed slowly with the pestle in order to determine the adsorption capacity of the solid carrier. The lowest amount of solid carrier capable of adsorbing the highest liquid SEDDS amount, resulting in free-flowing non-sticky powder, was chosen for further studies. Prepared powders were left to dry at room temperature and stored in a desiccator until further use.

##### Characterization of Solid SEDDS Containing LYS:SDS—Flow Properties

After determining the maximum oil loading capacity, solid SEDDSs were evaluated for their flow properties. Micromeritic properties of solid SEDDSs, such as bulk and tapped density, angle of repose, Carr’s index, and Hausner’s ratio, were determined.

The angle of repose was determined using the fixed funnel method. The funnel was fixed 2 cm above the flat surface and the fixed weight of the powder was released from the funnel until the top of the conical pile touched the end of the funnel. Using the height of the cone (*h*) and the radius of the flat base (*r*), the angle of repose (θ) was determined using Equation (8):(8)$$\theta ={tan}^{-1}\frac{h}{r}$$

The flow rate was determined as the ratio of mass (g) to the time needed for a powder to flow throughout the fixed funnel. Further, the known mass of solid SEDDS powders was placed in a 5 mL graduated cylinder, and bulk density (*ρ_b_*) was calculated from the ratio of apparent volume and weight. Tapped density (*ρ_t_*), as the ratio between tapped volume and weight, was determined after tapping the samples 1250 times using (Stampfvolumeter, JEL STAV 2003, J. Engelsmann AG, Ludwigshafen, Germany). Car index and Hausner ratio were calculated according to Equations (9) and (10):(9)$$CI\left(\%\right)=\frac{\rho_{t}-\rho_{b}}{\rho_{t}}\times 100$$
(10)$$Hausner\;ratio=\frac{\rho_{t}}{\rho_{b}}$$
where *ρ_b_* is bulk density and *ρ_t_* is tapped density.

##### Self-Emulsifying Properties of Solid SEDDS

To assess the self-emulsifying properties of the formulations after solidification, 100 mg of solid SEDDS powder was dispersed in distilled water (1:100) under 300 rpm stirring at 37 °C. The samples were further subjected to filtration (0.45 µm) to separate insoluble silica particles. Self-emulsification time, % transmittance, droplet size, polydispersity index, and zeta potential were evaluated. Organoleptic properties, such as phase separation and physical stability after dilution, were assessed visually. Droplet size and zeta potential of diluted solid SEDDS were evaluated using Malvern Nano ZS Zetasizer (Malvern Instruments Ltd., Malvern, UK).

##### Determination of LYS Activity after Incorporation into liquid and Solid SEDDS

After incorporation of LYS:SDS complex in liquid SEDDS preconcentrate, 100 µL of preconcentrate was emulsified in 1 mL of phosphate buffer, 6.8 pH at 37 °C, and 300 rpm (1:10 dilution). Liquid SEDDS loaded with LYS:SDS complex was adsorbed onto a solid carrier (1:2 ratio), and 100 mg of the prepared solid SEDDS was also emulsified in 1 mL of phosphate buffer, 6.8 pH at 37 °C, and 300 rpm (1:10 dilution). Then, 100 µL of each resulting emulsion was added to 2.5 mL of 0.25 mg/mL bacterial suspension of *M. lysodeikticus* and mixed with inversions for 20 s in a quartz cuvette. The activity of *LYS* was determined by measuring the degradation of a bacterial suspension at 450 nm every 5 s using a UV spectrometer (Genesys 10 S UV-VIS Spectrometer, Thermo Fisher Scientific Inc., Waltham, MA, USA). Enzyme activity (%) of LYS in SEDDS was compared to the activity of corresponding LYS:SDS complex as well as freshly prepared free LYS solution.

#### 2.2.5. Development and Characterization of Self-Emulsifying Tablets with LYS:SDS Complex

##### Compression of Tablets and Their Characterization

Next, 450 mg tablets were compressed at different compression forces using a Korsch EK0 (E. Korsch Maschienenfabrik, Berlin, Germany) single-punch tablet press with a 12 mm flat round punch. The final tablet formulation was composed of 35% solid SEDDS (1:1 ratio).

Obtained tablets were then characterized following Ph. Eur. monographs, regarding weight uniformity, diameter, thickness, hardness, friability, and disintegration time. The thicknesses and diameter were checked using a screw micrometer (Mitutoyo Co., Ltd., Kawasaki, Japan) with an accuracy of ±0.01 mm. The friability was assessed using an Erweka TA friability tester (Erweka GmbH, Heusenstamm, Germany). In a rotating drum (25 rpm), using previously weighted tablets, the test was run for 4 min. Then, tablets were brushed to remove excess powder and finally weighed. The percentage of weight loss was calculated as the difference between initial and final weight divided by initial weight. The hardness of tablets was checked using a Heberlein tablet hardness tester (Heberlein & Co. AG, Wattwil, Switzerland). The disintegration time was tested in distilled water at 37 ± 0.5 °C using an Erweka model ZT2 apparatus (Erweka GmbH, Heusenstamm, Germany).

##### Dissolution Studies

In vitro dissolution studies were conducted using the USP paddle method (Erweka DT 700, Erweka GmbH, Heusenstamm, Germany). Tablets were put into a 300 mL dissolution medium (PBS 6.8 pH) at 37 °C ± 0.5 °C and under 100 rpm stirring. At predetermined time intervals (5, 10, 20, 40, 60, 90, 120, 150, 180, 240, and 300 min), 3 mL of aliquot was withdrawn, filtered, and analyzed using a UV spectrophotometer (Spectronic Helios Alpha, 084304, Spectronic Unicam, Cambridge, UK) at 280 nm. Since the medium was not replaced throughout the experiment, the medium loss was taken into consideration when making calculations. The cumulative percentages of dissolved LYS from solid SEDDS were calculated and plotted versus time.

## 3. Results and Discussion

Considering the numerous challenges and risks involved in the development of oral drug delivery systems for biopharmaceuticals, the application of the QbD concept during their development and research is imperative. This concept not only shortens the development time but also ensures quality at every step of the development process, which is of great importance for the preservation of the biological activity of biopharmaceuticals during the manufacturing process [46]. It is worth noting that special attention must be paid in the case of biopharmaceuticals; thus, classical guidelines, which describe QbD (ICH Q82R, Q9, and Q10), must be extended with the ICH Q11 guideline. Additionally, the impacts of the CQAs on the QTPP are becoming progressively more challenging to comprehend due to risks connected with efficacy, immunogenicity, safety, pharmacokinetics, and complex technology transfer due to the sensitive nature of the biopharmaceuticals [46].

In our case, as for QTPP, the target final dosage form was a self-emulsifying tablet with LYS as a model biopharmaceutical. The LYS activity must be maintained during the entire development process. As formulation comprises three independent steps, we firstly identified CQAs for each development phase. A flowchart of the development process can be found in the Appendix A of this article (Appendix A).

An initial RA study was also performed for three independent development phases. Additionally, to show the cause-and-effect relationship between the material and process variables, Ishikawa diagrams were also constructed for all three formulation steps. Interdependence rating between CMAs/CPPs and CQAs for the complexation step can be seen in Table 3, while the corresponding Ishikawa diagram is shown in Figure 1. The RA results regarding the other two formulation steps can be found in the Appendix A of this article (Appendix A, Appendix A).

### 3.1. Preparation and Optimization of LYS:SDS HIP Complex

Hydrophobic ion pairing (HIP) was found to be a promising approach to enhance the encapsulation of hydrophilic biopharmaceuticals in lipid nanocarriers. In the present study, LYS, a 129-amino-acid protein with bacteriolytic properties, was used as a model API. Due to its net positive charge at physiological pH, LYS could be associated with anionic surfactants. Furthermore, sodium dodecyl sulphate (SDS) was chosen as the counterion, as it is well known that negatively charged sulphate groups, pKa (<0), in SDS can stoichiometrically bind LYS [39,40]. It was reported that many parameters, such as concentration, pH, ionic strength, and temperature, affect successful complexation between the counterion and hydrophilic macromolecule [13]. Thus, one of the first aims of this work was to quantify the effect of the most influential factors and their interactions on HIP complex formation and to obtain process design space (PDSs). As far as we know, this is the first study utilizing the QbD concept in HIP complex preparation.

To identify the most significant factors affecting the HIP of LYS with SDS, high-risk parameters resulting from the risk assessment study were firstly screened using a two-level five-factor fractional factorial design (2^5−2^). According to the results of the screening design (Equations (11) and (12); detailed statistical results can be found in Appendix A), three factors with the highest impact on responses, namely, molar ratio, pH, and temperature, were selected for the optimization design study. Mixing speed was fixed at a low level (300 rpm) as it was not found to be statistically significant for binding efficiency (y_1_) and had a negative coefficient (−7.09) in the case of enzyme activity (y_2_). Due to low values of coefficients and non-significance, mixing time was also excluded from the design and fixed at a lower level (15 min).
y_1_ = **75.57** + **13.67 x_1_** − 2.18 x_2_ − **2.30 x_5_**, (11)
with R^2^ = 0.98664, adjusted R^2^ = 0.97662, and mean square (MS) residuals = 5.34
y_2_ = **71.77** − **18.20 x_1_** + 4.51 x_2_ − 7.09 x_3_ − 5.41 x_5_, (12)
with R^2^ = 0.83843, adjusted R^2^ = 0.62301, and mean square (MS) residuals = 221.71

For further investigation, 2^3^ full factorial design was chosen. After adding experimentally obtained results of responses y_1_, y_2_ (Table 4.) to the design matrix, first-order polynomial equations were generated to quantify the effect of the independent variables on the responses. The best-fitting model was chosen based on the coefficient of determination R^2^, adjusted R^2^, and MS residuals value (detail statistical results can be found in Appendix A).

As expected, at a lower level (−1) of molar ratio, the binding efficiency was relatively low in comparison to values obtained at a high level (+1). Overall, the binding efficiency of LYS mostly depends on the molar ratio (Coeff. = 12.73), and it exhibited a linear relation with the investigated parameters (Equation (13)). pH value was not significant and the effect on the response (y_1_) was very small (Coeff. = 0.020), confirming that hydrophobic interactions, in addition the ion-pairing mechanism, are very important in the binding of LYS to anionic surfactants, which correlates with previously published results [39]. We assume that LYS tends to interact with the hydrophobic hydrocarbon tail of a surfactant via non-specific hydrophobic interactions, as the pH value approaches the pI value of LYS [39]. The effects of temperature as well as two- and three-factor interactions on binding efficiency of LYS were statistically neglectable (Equation (13)).
y_1_ = **74.73** + **12.73 x_1_** + 0.023 x_2_ + 0.048 x_3_ − 0.54 x_1_x_3_ − 0.21 x_2_x_3_ + 0.29 x_1_x_2_x_3_(13)
with R^2^ = 0.96138, adjusted R^2^ = 0.92275, and mean square residuals (MS) = 13.05; curvature (−7.66)

In the case of enzyme activity (Equation (14)), the factor with the highest coefficient was the molar ratio (Coeff. = −9.76), meaning that enzyme activity is lower in the case of a higher amount of anionic surfactant added in the complex preparation, as can be seen on the response surface plot (Figure 2b). Assuming that LYSs’ cationic sites are partially neutralized by ionic pairing with sulphate counterions, we anticipated that a higher counterion ratio would decrease enzyme activity and impede the binding of LYS to the anionic sites on bacteria’s cell walls that initiate cell lysis. A high positive coefficient was also observed in the case of pH value (Coeff. = +7.46), indicating higher enzyme activity at higher pH values, which is in line with previously published results [47]. The two-way interaction coefficient (x_1_x_2_) was also high (Coeff. = +7.78) and statistically significant, highlighting the role of molar ratio and pH interactions during the complexation process on enzyme activity.
y_2_ = **81.75** – **9.76 x_1_** + **7.46 x_2_** + 4.77 x_3_ + 7.78 x_1_x_2_ + 0.32 x_1_x_3_ + 1.30 x_2_x_3_
(14)
with R^2^ = 0.98673, adjusted R^2^ = 0.96461, and mean square residuals (MS) = 8.46; curvature (+3.96)

The response surface plots were generated for better visualization of the interaction effect between factors on outcomes (Figure 2). Pareto charts of the effects as well as correlation between the predicted and the observed values can be seen in the Appendix A.

To validate the obtained models, three replications of checkpoint formulations were prepared and investigated regarding binding efficiency and enzyme activity. It was shown that the design has good predictability, as minor differences between the predicted values and the average of experimental values were observed for both responses (Table 5).

Finally, the design space was made according to the recommendations of Appendix 2 of the ICH Q8 guidelines and following CQA acceptance criteria: binding efficiency >85% and enzyme activity >75% (Figure 3).

Based on the abovementioned results and design space, the optimal conditions for LYS:SDS-HIP complex preparation were identified as molar ratio 1:6, pH 9, and mixing speed 300 rpm, and they were, therefore, used for further studies. At these optimal conditions, the binding efficiency (%) and enzyme activity (%) were found to be 88.88 ± 0.05 and 77.19 ± 1.22, respectively. Furthermore, HIP of LYS with SDS resulted in a decreased aqueous solubility of LYS, allowing for its successful incorporation into the SEDDS.

### 3.2. Dissociation of LYS from the HIP Complex

To describe the nature of the interaction between LYS and counterion as well as the stability of HIPs, the dissociation of LYS from the HIP complex was examined. As it is known that SDS is a highly acidic counterion (pKa = −1.5) with a reported dissociation constant (K_d_) of 0.1 M [48], it is expected that it will be tightly bound to LYS, resulting in slower dissociation and, consequently, higher stability in the complex.

As can be seen in Figure 4, the dissociation percentage of LYS from the complexes after a period of 4 h for all formulations was very low (up to 10%). After 24 h, the highest dissociation percentage was 18%, proving high stability of the complex in the release medium (PBS pH 6.8). However, we assumed that dissociation is somewhat higher at 37 °C and under dynamic conditions (the dissociation study was performed at room temperature, without stirring the samples).

It is worth noting that the complex should have been entirely dissolved if ionic forces were the only factor that stabilized the HIP complex; thus, we anticipated that hydrophobic interactions are also very important in LYS:SDS complex stabilization, which correlates with other studies [49]. Thus, taking into account that hydrophobic interactions may also be involved in complex formation, due to the long hydrophobic chain of SDS, it can be concluded that the prepared complex is highly stable and lipophilic.

HIPs are known to have very low in vivo stability because of competing endogenous counterions that cause complete dissociation of the complex and, thus, as a result, low absorption membrane permeability [48]. Incorporating HIPs in lipid-based nanocarriers such as SEDDSs can provide sufficient protection and increase their in vivo stability. Although HIPs are reported to be in an associated form, or so-called “superassociated” form, following release from the lipid nanocarriers, due to time-dependent dissociation in the intestinal fluid, they can lose their capacity to permeate cell membranes to a greater extent [15]. This means that, as long as HIPs remain superassociated, they have more time to successfully permeate the absorption membrane. Recently, Shahzadi et al. highlighted the impact of superassociation on membrane permeability, reporting that superassociated HIPs showed a significant permeation-enhancing effect across Caco-2 cells (3.1-fold) and freshly excised rat intestinal mucosa (2.5-fold), compared to entirely dissociated HIPs [48].

These findings emphasize the significance of regulating HIP dissociation through the selection of an appropriate counterion with high stability, as well as the design of drug delivery systems capable of releasing HIPs in superassociated form at the absorption membrane.

### 3.3. Fourier-Transform Infrared Spectroscopy (FT-IR)

FT-IR spectra for all nine samples from the experimental design and corresponding physical mixtures (1:2, 1:6; LYS:SDS) were acquired to confirm complex formation and evaluate the secondary structure of LYS in the complex. In order to maintain its therapeutic activity, LYS must retain its native structure during the HIP process. It is common practice to utilize the amide I band of protein/peptide in FT-IR spectra (between 1600 cm^−1^ and 1700 cm^−1^) as an indicator for the secondary structure (backbone) conformation of protein/peptides [50]. As illustrated in Figure 5, amide I bands of the LYS:SDS complex and free LYS were comparable, suggesting that the enzyme maintains its secondary structure during the complexation process in all samples. The amide II (1558 cm^−1^) peak of LYS can also be well assigned in each case, confirming preserved secondary structure.

Two characteristic absorption bands corresponding to the SO_2_ stretch (1222 cm^−1^ asymmetric; 1085 cm^−1^ symmetric) were monitored in all samples to confirm successful binding between LYS and SDS. In the case of 1:2 and 1:6 physical mixtures, the presence of asymmetric stretch was confirmed at 1231 cm^−1^ and 1225 cm^−1^, respectively. The symmetric SO_2_ stretch is also visible at 1225 cm^−1^ and 1084 cm^−1^. However, we observed shifts in SO_2_ bands, asymmetric as well as symmetric ones, in all prepared complexes. Thus, this observation may be an indication of the successful formation of the complex between LYS and SDS.

The spectra of samples C1–C5, C3–C7, C2–C6, and C4–C8 (Table 1), which were complexed at different temperatures but with the same molar ratio and same pH value, were, likewise, found to be extremely comparable. This may serve as further evidence that the complexation temperature has very little effect on the complexations’ effectiveness, as we already found in our DOE study. It is also evident from the spectra that complex interactions between parameters during complexation play important roles in complexation efficiency.

### 3.4. Development and Characterization of Liquid-SEDDS Loaded with LYS:SDS Complex

Based on the literature review, as well as preliminary solubility studies, medium-chain triglycerides, polysorbate 80, and hydrophilic co-solvent PEG 400 were identified as suitable excipients for SEDDS formulation. All used excipients were safe (GRAS status) and were shown to readily diffuse through mucus [18].

Beyond choosing the excipients, it is essential to identify self-emulsification areas, i.e., optimal ratios of SEDDS components, to develop an optimized SEDDS formulation. This is usually performed based on traditional one-factor-at-a-time approaches through the construction of ternary phase diagrams or pseudo(ternary) phase diagrams. However, this approach is time-consuming, labor-intensive, and inadequate to analyze and quantify the effect of independent variables and their complex interactions on critical quality attributes of SEDDS [51]. Thus, in this study, we used a three-factor-constrained mixture of experimental design and response surface methodology to investigate individual and interaction effects of selected independent variables on SEDDS properties and optimize the liquid SEDDS for further complex incorporation studies.

Emulsification of liquid SEDDS preconcentrates led to a transparent dispersion with a high percentage of transmittance (%T) and small droplet size, except formulations L1 and L6, which showed a bluish appearance with higher droplet size, as listed in Table 6. The %T was, in most cases, >98%, which is an indication of a fast and reproducible emulsification process, except for samples L1 and L6, with %T of 3.14 ± 0.01% and 76 ± 0.01%, respectively. This decrease in %T is associated with increased oil concentration, an insufficient amount of surfactant, and a co-surfactant in the formulation to reduce the droplet size. According to [43] classification system, all formulations could be considered Grade I emulsions, except formulation L1, which can be labeled as a Grade II emulsion.

The droplet size of all examined SEDDSs was in the nanometer range, with narrow distribution (PDI between 0.09 and 0.4). As expected, it was observed that the droplet size increased from 11.2 nm to 177 nm when the concentration of lipid in preconcentrates increased from 5% to 40%. The surface charge of blank formulations was found to be negative (−5.6 mV to −21.10 mV) in all cases, which, according to previous studies, can be advantageous and result in higher mucus permeation ability, due to less interaction with the negatively charged cell membrane [52,53]. All blank emulsions showed no phase separation; they were all clear and homogenous after 7 days at 25 °C. Liquid preconcentrates and diluted samples did not exhibit phase separation when subjected to stress conditions (freeze/heat cycles), demonstrating the stability of the formulations.

The experimentally obtained values of responses, i.e., droplet size, polydispersity index, and zeta potential, were added to the statistical design matrix, and polynomial equations were generated. Based on the results of the responses, the software suggested a special cubic model as the best fitting for all the response variables. Two-dimensional contour plots were generated, providing a visualization of the influence of all three components in the mixture design on a chosen response (Appendix A).

Droplet size and polydispersity index increased with increasing oil content, as predicted and reported in numerous papers [54,55,56], and this relationship can also be seen from the obtained polynomial equations of the fitted model (Equations (15) and (16)) and corresponding positive coefficient values for all three components in the mixture design (detailed statistical results can be found in Appendix A). In addition, the interaction between oil ratio and surfactant content was significant, with a very high negative coefficient value (Coeff. = −279.52) on droplet size (y_3_). It is obvious from the negative values of the interaction effects (AB, ABC) that the surfactant works together with the co-surfactant to reduce the size of the oil droplets in the system; however, despite its high coefficient value, the interaction effect of ABC was found to be insignificant.
y_3_ = **177.96 A** + 10.35 B + 5.88 C − **279.52 AB** + 41.92 BC − 455.06 ABC (15)
with R^2^ = 0.9929 and adjusted R^2^ = 0.9811

In the case of polydispersity index (y_4_), the interaction between oil and surfactant content was also significant but with a positive value (Coeff. = +1.03), indicating direct proportionality between the independent and dependent variables (Equation (16)).
y_4_ = **0.35 A** + 0.09 B + **0.34 C** + **1.03 AB** −0.38AC + 0.13 BC − 2.40ABC, (16)
with R^2^ = 0.9817 and adjusted R^2^ = 0.9269

As for the effect of independent variables on zeta potential of the samples, the positive coefficient value of co-surfactant ratio (Coeff. = +13.49) implied that a higher co-surfactant concentration resulted in higher zeta potential (Equation (17)).
y_5_ = 10.81 A + 5.40 B + **13.49 C** − 30.81 BC(17)
with R^2^ = 0.3962 and adjusted R^2^ = 0.0339

Smaller droplet size facilitates drug absorption through biological membranes, and oil droplets with a negative charge can easily permeate the charged mucus layer, enhancing drug bioavailability. These properties were shown to be highly advantageous in the development of SEDDSs [52,53]. Thus, acceptance criteria for determination of the design space were droplet size smaller than 100 nm, PDI smaller than 0.3, and ZP higher than −3, as shown in Figure 6a–c.

The contour maps with the areas that fit the acceptance criteria of selected CQAs were overlapped into one graph to form a non-linear design space (marked with black lines) (Figure 6d). As design space revealed a broad range of possible mixtures meeting the set criteria in terms of droplet size, PDI, and zeta potential, it was taken into account that overall SEDDS properties, and published data up to now, suggested that formulations with a relatively high percentage of a hydrophilic co-surfactant are associated with a burst release of API [15]. It is, thus, usually advisable to use a lower hydrophilic co-surfactant percentage, if possible, to load API. Thus, three optimum formulations from the design space were chosen with a lower percentage of co-surfactant, and we tested the maximum payload of the complex to choose the optimum one.

### 3.5. Incorporation of the LYS:SDS Complex in Liquid SEDDS

On the basis of the obtained design space **(**Figure 6d) as well as literature-advisable percentage of hydrophilic co-surfactant, three formulations (L2, L9, L10) were chosen for further complex incorporation studies. After incorporation of the LYS:SDS complex, neither droplet size nor zeta potential changes significantly (*p* < 0.05), as can be seen in Table 7, thus indicating SEDDS stability. As expected, a slight increase in droplet size can be observed in all formulations loaded with LYS:SDS complex. All formulations showed an acceptable polydispersity index in a range of (0.21–0.32). SEDDSs loaded with LYS:SDS complex showed less negative zeta potential in comparison with blank SEDDS. A minor increase in droplet size was observed after storing samples at 25 °C for 7 days. Formulations showed constant stable negative zeta potential, which is a reliable indicator of emulsion stability (Table 7).

The optimal formulation was chosen according to the maximum payload of the complex in a corresponding liquid SEDDS, as indicated in Table 8. The maximum payload of the complex that could be dissolved in the preconcentrate was 4% (40 mg of the complex in 1 g preconcentrate), which corresponds to 3.57% LYS payload. These findings imply that the amount of complex that could be incorporated into formulations decreased as the oil concentration increased, which is likely related to the complex’s higher solubility in the utilized surfactant. Loaded SEDDS preconcentrates showed no signs of instability or drug precipitation after 48 h incubation at room temperature.

### 3.6. Robustness to Dilution and pH Changes

The potential effects of different dilution ratios, as well as stability under different pH conditions, were evaluated in vitro to prove SEDDS emulsion stability after oral administration and prevent burst release of the incorporated complex. The obtained results show that the chosen liquid SEDDS (L2) was robust to different dilution media and dilution ratios, as no significant difference (*p* < 0.05) in the droplet size and PDI was observed. Stability over the gastrointestinal pH range (1.4–6.8) was proven, as shown in Table 9.

### 3.7. Determination of log D_SEDDS/release medium_

The release behavior of HIPs from the SEDDS is crucial in their overall in vivo performance, and it is primarily controlled by the GIT environment of the body, rather than the formulation itself. SEDDSs can provide a protective effect for incorporated HIPs, but only if the complex remains in the oil droplets until it reaches the intestinal absorption barrier [57].

It is postulated that drug release from an SEDDS is a simple diffusion process from a lipophilic liquid phase of the SEDDS into an aqueous phase of release medium and can be explained using partition coefficient- *log D_SEDDS/release medium._* In order to reach the aqueous medium, the incorporated complex has to diffuse to the surface of the oil droplets and overcome a so-called interfacial barrier. Taking into account that the droplet size of the oil droplets is in the nano-sized range, the loaded complex can rapidly reach the surface of the droplets and release medium. Additionally, formulation components, such as polar solvents with high diffusion coefficients, may even accelerate complex diffusion and lower the interfacial barrier, causing release within seconds [45].

In the case of an SEDDS loaded with HIP complexes, release behavior is strongly impacted by dissociation of the complex in the release medium. Thus, determining maximum solubility of the complex in the release medium can be challenging. Dissociation of the complex over time, which is dependent upon pH, and ionic strength will cause the maximal drug solubility in the release medium to rise and, consequently, a decrease in *log D_SEDDS/release medium_*. Nazir et al. reported a significant decrease in *log D_SEDDS/release medium_* values and, consequently, faster release of HIPs with increasing pH and ionic strength in the release medium [58].

To improve the validity of *log D_SEDDS/release medium_* determination, maximum solubility of the LYS:SDS was determined in distilled water and PBS 6.8 (Table 10). Under simulated physiological conditions (PBS pH 6.8), *log D_SEDDS/PBS_* was relatively low (1.37), meaning that around 68% of the LYS will be immediately released in intestinal fluid from the SEDDS, assuming a dilution ratio of 1:50. However, dissociation of LYS in this release medium strongly impacts maximum solubility in the release medium and, thus, the partition coefficient. Therefore, considering that dissociation of the complex in distilled water is neglectable, *log D_SEDDS/DW_* of 2.72 seems to be more reliable. In that case, assuming the dilution of the SEDDS in the intestine is 1:50, a higher amount of drug will remain in the oily phase.

It was found in in vivo studies that complexes with higher log D values resulted in lower oral bioavailability. As a steep concentration gradient on the absorption membrane enhances drug uptake, a low log D seems to be favorable [45].

### 3.8. Preparation of solid SEDDS Containing LYS:SDS Complex

The optimal liquid SEDDS formulation was transformed into a solid SEDDS as a free-flowing powder using adsorption onto a solid carrier, as this technique is characterized by high lipid uptake (up to 80%) and content uniformity [59,60]. Therefore, choosing an appropriate solid carrier material was essential for creating an efficient solid SEDDS formulation. The most commonly used types of porous solid carriers are: micronized porous silica, fumed silica, magnesium-aluminometasilicates, silicon dioxide, calcium silicate, dibasic calcium phosphate, dextran, magnesium trisilicate, etc. [61].

Inorganic porous silicon dioxide (Syloid^®^ 244 FP and Syloid^®^ AL-1 FP) and magnesium-aluminometasilicates (Neusilin^®^ UFL2) were evaluated as potential solid carriers. In our preliminary studies, solid carriers were first screened for their oil adsorption capacity to determine appropriate solid carrier:SEDDS ratio and oil adsorption tendency. As illustrated in Appendix A, Neusilin^®^ UFL2 was found to have the highest adsorption capacity of 2:1. Syloid^®^ AL-1 FP was excluded due to unsatisfactory oil loading capacity. The oil adsorption tendency of Neusilin^®^UFL2 and Syloid^®^ 244 FP was found to be better over Syloid^®^ AL-1 FP; thus, micronized porous silica and aluminometasilicates were chosen for further studies.

The maximum solid carrier:liquid SEDDS ratio capable of maintaining dry powder was found to be 1:2 in the case of Neusilin^®^UFL2. The formation of either sticky powder or paste-like material was observed in the case of the 1:3 and 1:4 ratios (Neusilin^®^UFL2:liquid SEDDS), respectively.

Solid SEDDSs were further optimized based upon their micromeritic properties, as well as self-emulsification efficiency. According to the obtained results of flow properties (Table 11) of the solid SEDDS formulation, powder flow can be described as passable, according to European Pharmacopoeia, and poor for the Neusilin^®^UFL2 1:1 ratio. It was expected that the flowability of solid carriers will be impaired by the adsorption of the liquid SEDDS, which can be explained by an increase in the cohesiveness of the powders due to increased van der Waals forces and the formation of liquid bridges [59]. Therefore, the compaction of solid SEDDSs can be very challenging, as their behavior upon compression is poorly understood.

The prepared solid SEDDS emulsified quickly and formed transparent emulsions after dilution in distilled water. The obtained emulsion droplet size after dilution of the solid SEDDS with Neusilin^®^UFL2 as the solid carrier (1:2) in distilled water (1:100) exhibited a droplet size of 26.83 ± 2.78 nm, with PDI < 0.25 and slight negative zeta potential (−0.078 ± 0.071 mV), similar to the ones obtained from an unprocessed liquid SEDDS formulation, thus confirming the preserved self-emulsification properties of the solid SEDDS.

On the other hand, in the case of a solid SEDDS formulation with Syloid^®^ 244P as the solid carrier, it was not possible to obtain reliable size results using the dynamic light scattering technique, probably due to the interference of silica particles, which was also reported in other papers [62]. Additionally, although the droplet size was maintained in the nano size range, both solid SEDDSs revealed substantial variations in droplet size and PDI in different dilution media, such PBS (pH 6.8) and HCl (pH 1.4), compared to distilled water.

As Neusilin^®^UFL2 showed better self-emulsifying properties and higher oil adsorption capacity, it was chosen as a solid carrier for the preparation of a solid SEDDS loaded with the LYS:SDS complex.

### 3.9. Determination of LYS Activity after Incorporation in Liquid and Solid SEDDS

The enzyme activity of LYS after incorporation in the SEDDS was 80.46% (±3.45) and 69.31% (±2.97), corresponding to the activity of the lyophilized LYS:SDS complex and native LYS solution, respectively. Lower enzymatic activity of SEDDS in comparison with enzyme activity of LYS:SDS complex was expected, as higher affinity towards the oily phase limits the free access of the complex to interact with the substrate in the aqueous phase, which was also shown in previous studies in the case of trypsin and horseradish peroxidase [63,64]. After the solidification process, we also observed a further decrease in enzyme activity: 54.8% ± 8.4, 34.8 ± 5.3, and 29.7 ± 4.5 in comparison with liquid SEDDS, LYS:SDS complex activity, and native LYS, respectively.

### 3.10. Preparation of Self-Emulsifying Tablets Containing LYS:SDS Complex

After adsorption of the liquid SEDDS to the solid carrier, we obtained solid SEDDS powders, which were further compressed into tablets. Given the numerous drawbacks associated with incorporating liquid SEDDS into capsules, tablets were selected as the most acceptable and feasible solid dosage form. However, it must be pointed out that multiple challenges are observed during the preparation of self-emulsifying tablets [60]. Poor flow properties of a solid SEDDS lead to a higher final tablet weight, as other excipients must be added to the formulation to achieve acceptable compaction and compression characteristics.

Solid SEDDSs were mixed with the microcrystalline cellulose (Vivapur 102^®^) to ensure good compaction properties, and super-disintegrant croscarmellose sodium (Vivasol^®^) was added to not hinder the rapid self-emulsification of the solid SEDDS in dilution media.

In the scope of this study, Neusilin^®^UFL2 at a 1:1 ratio with 35% *w*/*w* SEDDS loading in the tablet was the only suitable one to produce self-emulsifying tablets with acceptable properties. In the case of Syloid^®^ 244FP, we were not able to obtain tablets with adequate hardness nor at a 1:1 ratio, which was also reported by other authors [65]. We assumed that these soft tablets are the result of ‘squeezing out‘ the adsorbed lipid components from pores of the solid carrier, and increased liquid load is associated with a decrease in tablet hardness.

Due to our awareness of the high final weight of the tablets, which may have an impact on patient compliance, 4.5 mg of the LYS:SDS complex load was achieved in tablets of approx. ∼450 mg. As this study can serve as a proof of concept, in this work, no additional efforts were made to load LYS at its therapeutic dose.

According to pharmacopeial methods, the prepared self-emulsifying tablets were assessed for standard quality control tests. The tablet weight variation (4.9%) was found to be within the pharmacopeial acceptance limits. The average diameter of prepared tablets was 12.07 ± 0.02 mm, and their thickness was 3.11 ± 0.10 mm. The tablet hardness was found to be 190.67 ± 5.03.

Friability was found to be 0.20 ± 0.16%, indicating good mechanical tablet strength, on which there were no visual defects. Disintegration time in distilled water was found to be 8.21 ± 0.17 min, thus adhering to the European Pharmacopoeia’s criteria for uncoated tablets.

### 3.11. Dissolution Studies

It was found in many studies that the release process is slowed down by the desorption of a liquid SEDDS from solid carrier surfaces and interparticulate voids of porous carriers [66]. The irreversible physiochemical interaction of liquid SEDDS excipients with the solid carriers can also explain the incomplete and impaired drug release [67].

Assuming that the LYS:SDS complex must firstly undergo desorption from the solid carrier and then LYS must dissociate from the complex, and keeping in mind that our complex dissociation studies revealed very slow dissociation of the LYS from the LYS:SDS complex (up to 10% in 4 h), we expected a very slow release of LYS from the final dosage form.

The results from the in vitro release study revealed that the total amount of the free LYS fraction released after 300 min was around 9.5%, as depicted in Figure 7.

Since it is hypothesized that liquid SEDDSs can be retained in the pores of solid carriers, the size, shape, and length of the pores, together with specific surface area, play important roles in the dissolving behavior of these systems [68]. Neusilin^®^UFL2, which was used as a solid carrier in this study, has a large surface area (350.33 ± 2.88 m^2^/g; particle size: 3.1 µm) and very large pores (the size of mesopores: 7.62 nm, micropores: 0.45 nm, pore volume: 0.7 cm^3^/g) [69]; thus, we assumed that liquid SEDDSs filled the intraparticular pores of the adsorbent. Therefore, we predicted that a liquid SEDDS would be entrapped in the solid carrier’s pores, impeding contact with the dissolution media. Williams et al. also stated that the composition of SEDDS and surfactant amount can also impact the penetration of dilution media into pores and, thus, impede in vitro dissolution [70].

## 4. Conclusions

In this study, the potential of combining the benefits of solid dosage forms and liquid nanocarriers (SEDDSs) that could be of particular interest for the oral delivery of biopharmaceuticals was illustrated. Hydrophobic ion pairing with an anionic surfactant, SDS, sufficiently increased the lipophilicity of LYS, thus making it suitable for incorporation into liquid SEDDSs. After choosing an adequate solid carrier, liquid SEDDSs can easily be transformed into solid SEDDSs, thus making the formulation more stable and convenient for patients. An enzyme retains its biological activity during the whole development process, thus proving that self-emulsifying lipid nanocarriers can serve as promising carriers for oral delivery of biopharmaceuticals.

The use of the QbD concept during development provided some new insights and helped quantify complex interactions between material and process variables on critical quality attributes of the formulation. We believe that QbD-guided development of SEDDSs carrying biopharmaceuticals together with the development of reliable and predictive in vitro characterization techniques could bridge the gaps between academia and the pharmaceutical industry in this field and, thus, successfully fulfil therapeutic needs. Detailed studies that focus on the cytocompatibility of the formulation, as well as comprehensive in vivo studies in animal models, are the next steps in paving the way to this promising approach.

## Figures and Tables

**Figure 1 pharmaceutics-15-00995-f001:**
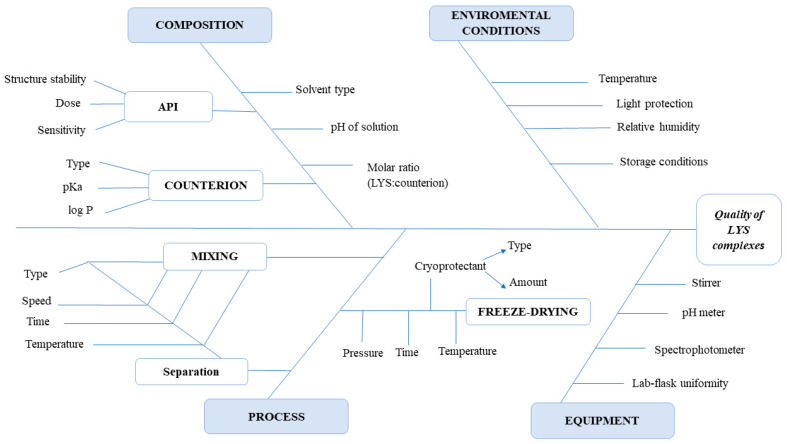
Ishikawa quality diagram of LYS HIP complex.

**Figure 2 pharmaceutics-15-00995-f002:**
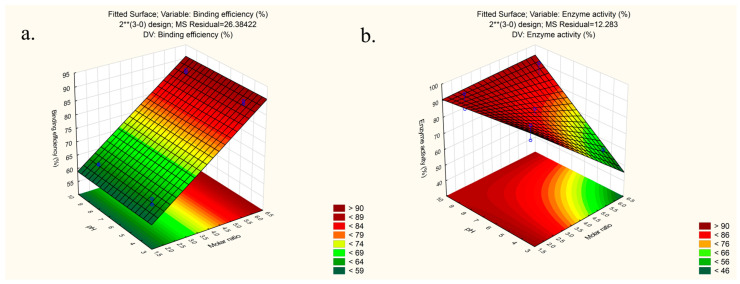
The response surfaces (temperature at zero level −31 °C) show the effect of the significant examined variables on the (**a**) binding efficiency and (**b**) enzyme activity.

**Figure 3 pharmaceutics-15-00995-f003:**
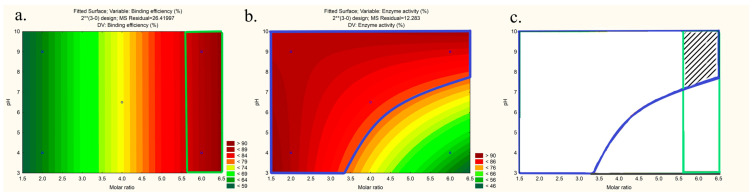
Contour plots of the (**a**) binding efficiency and (**b**) enzyme activity at 31 °C with acceptance criteria, and (**c**) illustration of the design space (marked area indicates the process design space).

**Figure 4 pharmaceutics-15-00995-f004:**
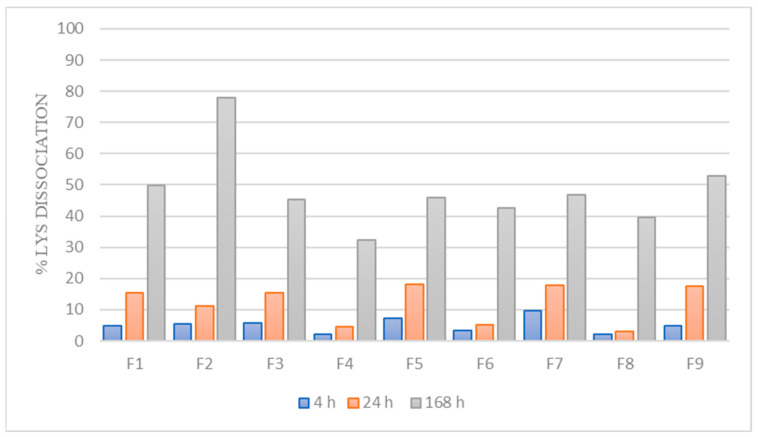
Dissociation of LYS from the complexes after incubation in PBS (6.8 pH) for 4 h, 24 h, and 168 h.

**Figure 5 pharmaceutics-15-00995-f005:**
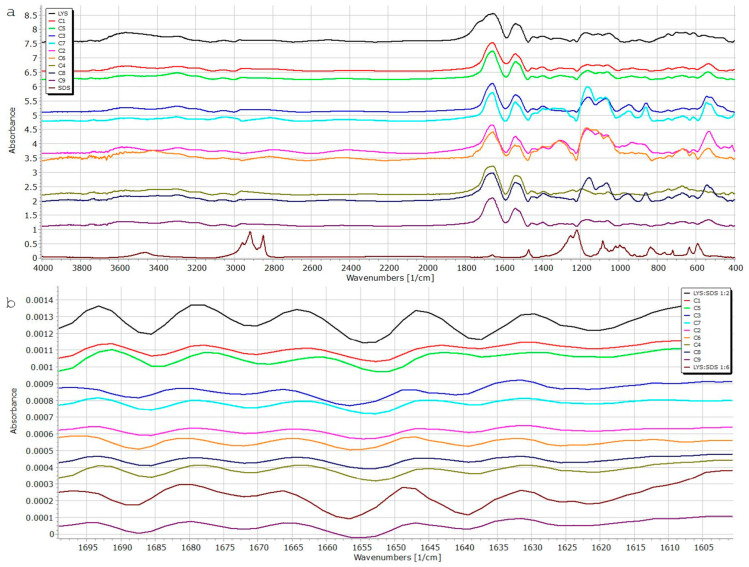
FT-IR spectra of the LYS:SDS complexes (**a**) and the second derivative of the amide I bands (**b**).

**Figure 6 pharmaceutics-15-00995-f006:**
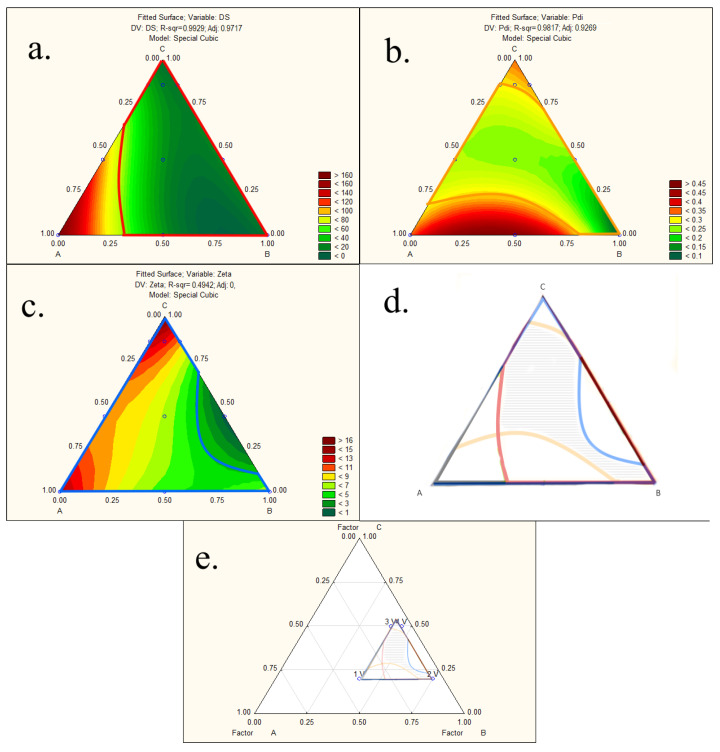
Two-dimensional contour plots of the (**a**) droplet size, (**b**) PDI, and (**c**) zeta potential with acceptance criteria (droplet size—red, PDI—orange, zeta potential—blue) and (**d**,**e**) illustration of the design space (marked area indicates the process design space).

**Figure 7 pharmaceutics-15-00995-f007:**
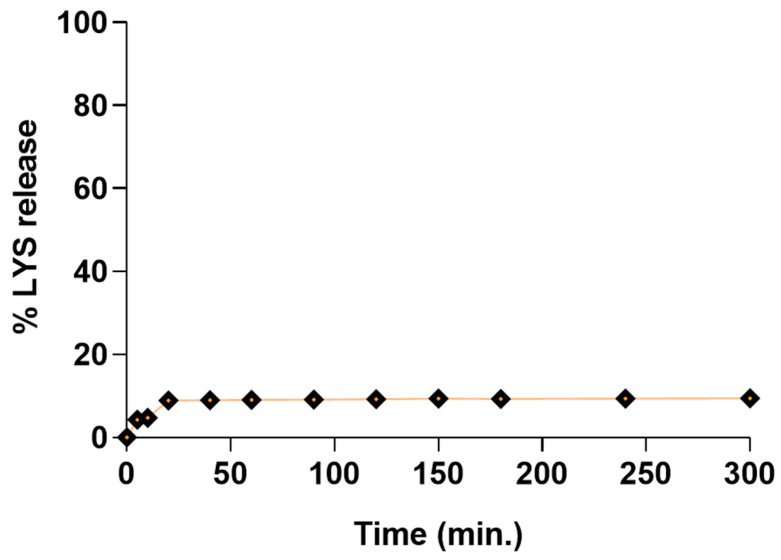
The dissolution profile of LYS from self-emulsifying tablets in PBS (6.8 pH). Solid SEDDSs were composed of liquid SEDDS adsorbed onto Neusilin^®^UFL2 at a weight ratio of 1:1. Each value designates the mean ± S.D. (*n* = 3).

**Table 1 pharmaceutics-15-00995-t001:** Design matrix for full factorial design with three factors, at two levels with one central point.

Experiment (C1–C9)	Molar ratio (LYS:SDS) (x_1_)	pH (x_2_)	Temperature (°C) (x_3_)
1	1:2 (−1)	4 (−1)	25.0 (−1)
2	1:6 (+1)	4 (−1)	25.0 (−1)
3	1:2 (−1)	9 (+1)	25.0 (−1)
4	1:6 (+1)	9 (+1)	25.0 (−1)
5	1:2 (−1)	4 (−1)	37.0 (+1)
6	1:6 (+1)	4 (−1)	37.0 (+1)
7	1:2 (−1)	9 (+1)	37.0 (+1)
8	1:6 (+1)	9 (+1)	37.0 (+1)
9	1:4 (0)	6.5 (0)	31.0 (0)

**Table 2 pharmaceutics-15-00995-t002:** Composition (%, *w*/*w*) of liquid preconcentrates according to the mixture design experimental layout.

Formulation	Oil	Surfactant	Co-Surfactant	Oil:S_mix_	S:CS
L1	40.0	40.0	20.0	1:1.5	2:1
L2	5.0	75.0	20.0	1:19	4:1
L3	10.0	40.0	50.0	1:9	1:1.3
L4	5.0	45.0	50.0	1:19	1:1
L5	5.0	60.0	35.0	1:19	1:1.7
L6	25.0	40.0	35.0	1:3	1:1.4
L7	22.5	57.5	20.0	1:3.4	1:2.9
L8	7.5	42.5	50.0	1:12.3	1:1.2
L9	15.0	50.0	35.0	1:5.7	1:1.4

**Table 3 pharmaceutics-15-00995-t003:** Interdependence rating between CMAs/CPPs and CQAs as for the complexation step (red color—high dependence, yellow—medium dependence, green—low dependence).

CQA	CMAs/CPPs							
	Type of Counterion	Molar Ratio (LYS:Counterion)	pH of the Solution	Ionic Strength of Solution	Mixing Type	Mixing Speed	Mixing Temperature	Centrifugation Speed
Binding efficiency	High	High	High	High	Medium	Medium	Medium	Low
Enzyme activity	High	Medium	Medium	Medium	High	High	High	High
Zeta potential	High	High	High	Medium	Low	Low	Low	Low
Dissociation of the complex	High	High	High	High	Low	Low	Low	Low
Lipophilicity of the complex	High	High	Low	Low	Low	Low	Low	Low
HIP payload	High	High	High	Medium	Low	Low	Low	Medium

**Table 4 pharmaceutics-15-00995-t004:** Binding efficiency and enzyme activity results (2^3^ full factorial design).

Sample	Binding Efficiency (%) (y_1_)	Enzyme Activity (%) (y_2_)
1	61.77	87.10
2	88.83	53.70
3	62.77	86.14
4	88.77	79.25
5	63.93	95.70
6	87.71	58.95
7	62.94	95.35
8	87.94	94.34
9	67.92	85.27

**Table 5 pharmaceutics-15-00995-t005:** The experimental and predicted values of the responses of the optimized complex LYS:SDS. Data are presented as mean ± SD (*n* = 3).

Responses	Check Point Formulation	Factors			Experimental Value	Predicted Value	% Bias
		x_1_	x_2_	x_3_			
Binding efficiency (%)	1	1:6	9	25	88.88 ± 0.05	87.91	0.48
	2	1:6	6.5	25	86.43 ± 0.77	87.94	0.755
Enzyme activity (%)	1	1:6	9	25	77.19 ± 1.22	80.84	1.82
	2	1:6	6.5	25	59.65 ± 1.61	66.90	3.62

**Table 6 pharmaceutics-15-00995-t006:** Characterization of blank SEDDS after dilution (1:100) in distilled water.

Code	Appearance	SE Time (min)	% T	Droplet Size (nm)	PdI	Zeta Potential (mV)
L1	bluish	≤2	3.14 ± 0.01	177.20 ±1.56	0.348 ± 0.03	−13.40 ± 0.20
L2	translucent	≤1	99.43 ± 0.23	11.22 ± 0.14	0.095 ± 0.02	−6.96 ± 2.81
L3	translucent	≤1	98.93 ± 0.13	20.40 ± 0.08	0.289 ±0.014	−7.39 ± 0.42
L4	translucent	≤1	99.37 ± 0.12	14.53 ± 0.42	0.301 ± 0.01	−5.70 ± 0.58
L5	translucent	≤1	99.43 ±0.03	15.87 ± 0.67	0.239 ± 0.011	−0.73 ± 0.55
L6	slightly bluish	≤1	76.00 ±0.01	107.47 ± 1.26	0.250 ± 0.004	−10.29 ± 0.64
L7	translucent	≤1	99.55 ± 0.08	24.43 ± 0.82	0.481 ± 0.016	−5.62 ± 0.47
L8	translucent	≤1	99.67 ± 0.09	25.09 ± 1.25	0.336 ± 0.033	−21.10 ± 1.95
L9	translucent	≤1	99.55 ± 0.08	21.79 ± 0.14	0.239 ± 0.003	−5.56 ± 1.62

Data shown as mean ± SD (*n* = 3). SE-Self-emulsification time.

**Table 7 pharmaceutics-15-00995-t007:** Characterization of LYS:SDS-complex-loaded SEDDS after dilution (1:100) in distilled water 24 h after dilution and after 7 days.

Code	Appearance	SE Time (min)	% T	Droplet Size (nm)	PdI	Zeta Potential (mV)
After 24 h						
L2	translucent	≤20 s	99.71 ± 0.04	13.02 ± 0.54	0.245 ± 0.008	−4.85 ± 0.50
L9	translucent	≤20 s	97.98 ± 0.17	24.98 ± 0.35	0.320 ± 0.020	−5.56 ± 1.53
L10	translucent	≤20 s	99.48 ± 0.11	14.62 ± 0.19	0.214 ± 0.037	−3.09 ± 0.39
After 7 days						
L2	translucent	≤20 s	99.34 ± 0.05	13.84 ± 0.74	0.218 ± 0.003	−0.49 ± 0.41
L9	translucent	≤20 s	97.56 ± 0.15	29.08 ± 0.54	0.210 ± 0.005	−9.60 ± 3.30
L10	translucent	≤20 s	99.45 ± 0.12	15.12 ± 0.23	0.191 ± 0.018	−5.81 ± 3.50

Data shown as mean ± SD (*n* = 3).

**Table 8 pharmaceutics-15-00995-t008:** The maximum payload of the LYS:SDS complex in liquid SEDDS.

Code	Oil (%)	Surfactant (%)	Co- Surfactant (%)	Maximum Concentration of Dissolved Complex (%)
L2	5.0	75.0	20.0	4.0
L9	15.0	50.0	35.0	1.0
L10	10.0	62.5	27.5	2.0

**Table 9 pharmaceutics-15-00995-t009:** Robustness to dilution and pH changes for L2 formulation.

Medium	Droplet Size (nm)	PDI	Droplet Size (nm)	PDI
	1:100		1:1000	
DW	13.02 ± 0.54	0.245 ± 0.008	21.31 ± 2.50	0.135 ± 0.007
PBS pH 6.8	14.13 ± 0.35	0.223 ± 0.017	18.02 ± 0.94	0.418 ± 0.029
0.1 M HCl pH 1.4	13.94 ± 0.52	0.208 ± 0.001	14.83 ± 0.19	0.317 ± 0.032

Data are shown as mean ± SD (*n* = 3).

**Table 10 pharmaceutics-15-00995-t010:** log D_SEDDS/release medium_.

Release Medium	Log D	C Oil Droplets (%)	C in RM (%)
PBS 6.8 pH	1.37	31.80	68.19
DW	2.72	91.29	8.71

V(SEDDS) = 1 mL, V (RM) = 50 mL.

**Table 11 pharmaceutics-15-00995-t011:** Micromeritic properties of prepared solid SEDDS.

Solid Carrier (SC)	Ratio (SC-SEDDS)	Bulk Density (g/mL)	Tapped Density (g/mL)	Carr Index (%)	Hausner Ratio	Angle of Repose (°)	Flow Time (s)
Neusilin^®^UFL2	1:1	0.2056	0.2977	30.95	1.448	44.29	16
	1:2	0.3246	0.4253	23.68	1.310	42.73	9
Syloid^®^ 244P	1:1	0.1363	0.1828	25.42	1.341	45.51	8
	1:2	0.2953	0.3937	25.00	1.333	44.08	7

## Data Availability

Not applicable.

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
