# Peer review of "Quality by Design-Based Development of Solid Self-Emulsifying Drug Delivery System (SEDDS) as a Potential Carrier for Oral Delivery of Lysozyme"

_pharmaceutics, 2023, doi:10.3390/pharmaceutics15030995_

Round 1

Reviewer 1 Report

I review the manuscript entitled "Quality by Design based development of solid self-emulsifying drug delivery system (SEDDS) as a potential carrier for oral  delivery of lysozyme, thoroughly and found it suitable for publication in pharmaceutics journal. The MS is well written and appropriate. All the sections have been properly designed and well explained. The authors have conducted a tremendous piece of work. Few points are suggested which will further improve the manuscript. 1- why the LYS was used as a model drug? 2- Which characterizations were performed for solidified SEDDS (powdered). 3. Explain the stability parameters in abstract. 4. Add some introduction of Lysozyme in the Introduction part, similarly briefly explain QbD design. 5- It will be better if the graphs for the Zeta analysis are pasted in the MS along with the table, 6- Add some statistics to the results, i didnt find any statistics significance in the results, atleast explain in the abstract. 

Author Response

Response to Reviewer 1:

First of all, we woul like to thank the valuable comments of the Reviewer which helped to improve the quality of our Manuscript.

Reviewer #1 had six questions/suggestions, and we would like to explain point-by-point the details of the revisions in the manuscript and our responses to the reviewer's comments. The revised parts of the manuscripts for this reviewer are marked with red color.

  1. “Why the LYS was used as a model drug?”

However, the main effects of lysozyme are well known since almost a century, according to recent studies lysozyme can justify the title "enzyme of the future" since beside of its well-known antibacterial, antiviral, antifungal activity anti-inflammatory, anticancer, and immunomodulatory action was also discovered, which puts the this enzyme becoming the center of interest of many scientists again. It would be suitable to formulate it in the oral dosage form with adequate bioavailability after oral administration. As secondary reason it could be mentioned that it is relatively cheap and easily accessible, which makes it ideal candidate for our studies, because the equipment used for the processing into solid dosage forms requires the use of high amount of materials.

Furthermore, after a comprehensive literature review, we didn’t find any work in which lysozyme was used as a model drug for SEDDS, so we thought it can be suitable to try this enzyme, which is highly dependent on its tertiary structure for keeping biological activity and its activity can be relatively easily measured. As we planned four phases of experimental work (complexation of a model drug, liquid SEDDS, solid SEDDS, and tablet as final dosage form), it was very important to have information about LYS activity during all development phases, to prove that enzyme can retain its biological activity, despite manufacturing process. The QbD methodology helped us to gain mathematical equations, that deeply explain relationships between formulation and process parameters and enzyme activity. Thus, after developing the process design space (PDS) we can choose the most suitable formulation and process parameters that will lead to formulation with the highest enzyme activity.

  1. “Which characterizations were performed for solidified SEDDS (powdered)”

As it was summarized in Table 11. the solidified SEEDs formulation went through on a full powder rheological characterization process (flow properties, angle of repose) and compactibility (bulk and tapped density, Carr’s index, and Hausner’s ratio) were investigated). The powdered samples were also dispersed in water and the same properties as for liquid SEDDSs (self-emulsification time, % transmittance, droplet size, polydispersity index, and zeta potential) were evaluated after emulsification of solid SEDDS as well as the LYS activity was determined prior to tablet compression as it is discussed in lines 895-906.

  1. “Explain the stability parameters in abstract”

We agree with the suggestion, so we added stability parameters to the abstract as follows:

“The final formulation of liquid SEDDS carrying LYS:SDS complex showed satisfactory in vitro characteristics as well as self-emulsifying properties (droplet size: 13.02 nm, PDI: 0.245, and zeta potential: -4.85 mV). The obtained nanoemulsions were robust to dilution in different media and highly stable after 7 days with a minor increase in droplet size (13.84 nm) and constant negative zeta potential (-0.49 mV)”

  1. “Add some introduction of Lysozyme in the Introduction part, similarly briefly explain QbD design”

Following the recommendations of the reviewers, we have broadened the Introduction part, explaining lysozyme as our model drug:

“From its discovery by A. Fleming in 1921, LYS (or muramidase or N-acetylmuramic acid hydrolase E.C. 3.2.1.17) attracts attention as an effective antimicrobial agent. Through the hydrolysis of the 1,4-glycosidic bonds between N-acetylglucosamide (NAG) and N-acetylmuramic acid (NAM) in the polysaccharide backbone of the peptidoglycans of the Gram-positive bacterial cell wall, lysozyme exerts its antimicrobial activity [1]. This basic protein is composed of 129 amino acids and is naturally presented in the human body (saliva, tears, mucus) as well as tissues of animals and plants.”

We added a short explanation of QbD in the introduction part, as we already discussed some most important issues of QbD in Section 2.2.1. and at the beginning of Section 3.

“The QbD concept is a knowledge and risk assessment-based quality management approach, used in pharmaceutical research and development. The QbD or the “GMP of the 21st century” is the methodology where the product and the manufacturing process are designed and developed according to the previously defined expectations [36]. This holistic and systemic approach will be used in all parts of this work, in order to identify all risks as well as critical material attributes and process parameters whose variability has a critical effect on the aimed product quality.”

We also provided an additional explanation of DOE results in methods, to avoid confusion regarding statistical analyzes.

  1. “It will be better if the graphs for the Zeta analysis are pasted in the MS along with the table”,

Due to relatively high number of samples and the size of the raw graphs, to paste them into the manuscript would unnecessarily increase the size of the paper without the giving of additional information on the results. Nevertheless, we are pasting some of the graphs about the results pf the results of the LYS:SDS complex loaded SEDDS after dilution (1:100) in distilled water 24h after dilution:

Formulation: L2

Formulation L9

Formulation L10

  1. “Add some statistics to the results, I didn’t find any statistics significance in the results, at least explain in the abstract.” 

As all experimental phases were done according to the QbD methodology, we used Design of Experiments (DOE) as a statistical method used to determine the effect of multiple variables on an outcome. The goal was to identify which variables have the most significant impact on the outcome, and to optimize the values of those variables in order to achieve the desired outcome. After conducting a DOE analysis, statistical equations are often generated to summarize the results. These equations can be used to predict the outcome based on the values of the input variables. The equations typically take the form of a mathematical model, which may include linear or nonlinear terms, as well as interaction terms between the variables (ex. y = b0 + b1x1 + b2x2)

The coefficients of the factors (b1, b2) describe the change in the CQA if the factor value is increased from the 0 to the +1 level, with calculation on the basis of linear regression. Used confidence interval was 95% and the alpha value indicating significant factors was 0.05.

If the influence was statistically significant, the factor in the equation is highlighted in bold. For example in case of our equation 14:

y2= 81.75 - 9,76 x1 + 7.46 x2 + 4.77 x3 + 7.78 x1x2 + 0.32 x1x3 + 1.30 x2x3                 

with R2= 0.98673, adjusted R2= 0.96461, and Mean square Residuals (MS) = 8.46; Curvature (+3.96)

effect of factor x1, x2 and their interaction was statistically significant (p<0.05). In case of enzyme activity the factor with the highest coefficient was the molar ratio (Coeff. =-9.76), meaning that enzyme activity is lower in the case of a higher amount of anionic surfactant added in complex preparation; The two-way interaction coefficient (x1x2) was also high (Coeff.=+7.78) and statistically significant, highlighting the role of molar ratio and pH interactions during the complexation process on enzyme activity.

Here is the table with detailed statistics, that we used to generate the equation 14.:

Factor

Effect Estimates; Var.:Enzyme activity (%); R-sqr=,98673; Adj:,96461 (Design: 2**(3-0) design ([No active dataset]) in Workbook3)
2**(3-0) design; MS Residual=8,458545
DV: Enzyme activity (%)

Effect

Std.Err.

t(2)

p

-95,% (Cnf.Limt)

+95,% (Cnf.Limt)

Coeff.

Std.Err. (Coeff.)

-95,% (Cnf.Limt)

+95,% (Cnf.Limt)

Mean/Interc.

81.7518

1.1682372

69.978749

0.00020414

76.7253

86.7782968

81.7517778

1.1682372

76.7253

86.7782968

(1)Molar ratio

-19.5163

2.47820534

-7.8751545

0.01574454

-30.1791

-8.853393

-9.758125

1.23910267

-15.0896

-4.4266965

1 by 2

15.5653

2.47820534

6.28085565

0.02442422

4.9024

26.228107

7.782625

1.23910267

2.4512

13.1140535

(2)pH

14.9113

2.47820534

6.016955

0.02652729

4.2484

25.574107

7.455625

1.23910267

2.1242

12.7870535

(3)Temperature

9.5423

2.47820534

3.85046786

0.06131099

-1.1206

20.205107

4.771125

1.23910267

-0.5603

10.1025535

2 by 3

2.6098

2.47820534

1.05308061

0.4027547

-8.0531

13.272607

1.304875

1.23910267

-4.0266

6.63630348

1 by 3

0.6363

2.47820534

0.25673821

0.82137825

-10.0266

11.299107

0.318125

1.23910267

-5.0133

5.64955348

We provided additional explanation of DOE results in methods (2.2.1.) and according to suggestion of other reviewer we will add detail statistics in supplementary material.

“After conducting a DOE analysis, statistical equations are often generated to summarize the results. The equations can be used to predict the outcome (y) for any combination of input variable values (x1 and x2). The coefficients of the factors describe the change in the CQA if the factor value is increased from the 0 to the +1 level, with calculation on the basis of linear regression. The confidence interval was 95% and the alpha value indicating significant factors was 0.05.”

Reviewer 2 Report

The authors have presented an interesting study on formulating lysozyme in solid self-emulsyfing drug delivery systems.

The paper is well organized, however I do have some suggestions for further improvement.

Please discuss in more detail the biopharmaceutical properties of lysozyme, especially in the context of selection SEDDS. Also, are there any other reported formulation approaches to oral delivery of lysozyme. Take into account the loading capacity of delivery systems.

How were the levels for variation of experimental factors (Table 1) selected?

Why was the risk for dissociation of the complex classified as low regarding the type of counter-ion? 

Please provide detailed statistical analysis of mathematical models obtained through experimental design in the supplementary file.

My major concern for this study are the results of the dissolution study. What about different experimental conditions, addition of surfactants? What about in vitro lipolysis and/or digestion study? One of the main reasons for the formulation development was the drug delivery, yet the system has failed to be demonstrated as efficient. Please assess this issue.

Author Response

Response to Reviewer 2:

First of all, we woul like to thank the valuable comments of the Reviewer which helped to improve the quality of our Manuscript.

Reviewer #2 had five questions/suggestions, and the revised parts of the manuscripts for this reviewer are marked with blue color.

  1. “Please discuss in more detail the biopharmaceutical properties of lysozyme, especially in the context of selection SEDDS. Also, are there any other reported formulation approaches to oral delivery of lysozyme. Take into account the loading capacity of delivery systems.”

As oral delivery of protein drugs remained one of the most challenging approaches in formulation development, we were interested in solid SEDDS as potential carriers for oral drug delivery. After a comprehensive literature review, we didn’t find any work in which lysozyme was used as a model drug for SEDDS, so we thought it can be suitable to try this enzyme, which is highly dependent on its tertiary structure for keeping biological activity and its activity can be relatively easily measured. We were primarily focused on performing QbD concept in formulation development, to quantify complex interactions between material and process variables on critical quality attributes of these formulations.

One of the major limitations of SEDDS is loading capacity, although recently published works suggest that even very high loadings can be achieved. We were aware of the fact that it will be challenging to load lysozyme in its therapeutic dose (30-90 mg if it used for antibacterial purposes), so this was not our primary goal in this research. We wanted to make proof of the concept, that proteins can be formulated in solid SEDDS without losing their biological activity and highlight the importance of using QbD-guided development in formulating such systems.

According to rewiever suggestion, we have discussed biopharmaceutical properties of lysozyme in introduction part:

“LYS is available on the market as a conventional tablet and syrup, however, there is limited data available on its pharmacokinetics in humans. Studies have suggested that it is absorbed from the intestine in humans, although the extent of absorption is not as high [2–4]. It was found that lysozyme absorption in the gut was segment-specific and that it was absorbed preferentially from the upper part of the intestine, most likely through clathrin-mediated endocytosis [2]. Therefore, LYS  formulation in a stable oral solid dosage form with adequate bioavailability may contribute to managing many diseases.”

  1. “How were the levels for variation of experimental factors (Table 1) selected?”

We have chosen the levels following an extensive literature review, on the hydrophobic ion pairing of LYS.

  1. “Why was the risk for dissociation of the complex classified as low regarding the type of counter-ion?”

Thank you very much for this question. The risk for dissociation of the complex should be classified as high regarding the type of counter-ion. We corrected this mistake in the manuscript. 

  1. “Please provide detailed statistical analysis of mathematical models obtained through experimental design in the supplementary file.”

We have added a detailed statistical analysis of mathematical models in the supplementary file.

Table S3. Two-level five-factor fractional factorial design (25–2)-detail statistical results

Factor

Effect Estimates; Var.:BE %; R-sqr=,98664; Adj:,97662 (Design: 2**(5-2) design (Spreadsheet1) in Workbook2) 2**(5-2) design; MS Residual=5,340902
DV: BE %

Effect

Std.Err.

t(4)

p

-95,% (Cnf.Limt)

+95,% (Cnf.Limt)

Coeff.

Std.Err. (Coeff.)

-95,% (Cnf.Limt)

+95,% (Cnf.Limt)

Mean/Interc.

75.56912

0.817076

92.48729

0.000000

73.30055

77.83768

75.56912

0.817076

73.30055

77.83768

(1)Molar ratio

27.35984

1.634151

16.74253

0.000075

22.82271

31.89697

13.67992

0.817076

11.41135

15.94848

(5)Temperature

-4.60709

1.634151

-2.81925

0.047864

-9.14422

-0.06996

-2.30354

0.817076

-4.57211

-0.03498

(2)pH

-4.37424

1.634151

-2.67676

0.055415

-8.91137

0.16289

-2.18712

0.817076

-4.45568

0.08145

Factor

Effect Estimates; Var.:Enzyme activity; R-sqr=,83843; Adj:,62301 (Design: 2**(5-2) design)2**(5-2) design; MS Residual=221,7099
DV: Enzyme activity

Effect

Std.Err.

t(3)

p

-95,% (Cnf.Limt)

+95,% (Cnf.Limt)

Coeff.

Std.Err. (Coeff.)

-95,% (Cnf.Limt)

+95,% (Cnf.Limt)

Mean/Interc.

71,7775

5,26438

13,63455

0,000853

55,0239

88,53112

71,7775

5,264384

55,0239

88,53112

(1)Molar ratio

-36,4150

10,52877

-3,45862

0,040681

-69,9222

-2,90776

-18,2075

5,264384

-34,9611

-1,45388

(3)Mixing speed

-14,1800

10,52877

-1,34679

0,270755

-47,6872

19,32724

-7,0900

5,264384

-23,8436

9,66362

(5)Temperature

-10,8200

10,52877

-1,02766

0,379722

-44,3272

22,68724

-5,4100

5,264384

-22,1636

11,34362

(2)pH

9,0350

10,52877

0,85813

0,453915

-24,4722

42,54224

4,5175

5,264384

-12,2361

21,27112

Table S4. 23 full factorial design -detail statistical results

Factor

Effect Estimates; Var.:Binding efficiency (%); R-sqr=,96138; Adj:,92275 (Design: 2**(3-0) design 2**(3-0) design; MS Residual=13,05386
DV: Binding efficiency (%)

Effect

Std.Err.

t(2)

p

-95,% (Cnf.Limt)

+95,% (Cnf.Limt)

Coeff.

Std.Err. (Coeff.)

-95,% (Cnf.Limt)

+95,% (Cnf.Limt)

Mean/Interc.

74.73111

1.7028

43.88642

0.0005

67.4044251

82.05780

74.73111

1.7028

67.40443

82.05780

(1)Molar ratio

25.46000

3.6122

7.04824

0.0195

9.91775181

41.00225

12.73000

1.8061

4.95888

20.50112

1 by 3

-1.07000

3.6122

-0.29621

0.7950

-16.6122482

14.47225

-0.53500

1.8061

-8.30612

7.23612

1*2*3

0.57000

3.6122

0.15780

0.8891

-14.9722482

16.11225

0.28500

1.8061

-7.48612

8.05612

2 by 3

-0.42500

3.6122

-0.11766

0.9171

-15.9672482

15.11725

-0.21250

1.8061

-7.98362

7.55862

(3)Temperature

0.09500

3.6122

0.02630

0.9814

-15.4472482

15.63725

0.04750

1.8061

-7.72362

7.81862

(2)pH

0.04500

3.6122

0.01246

0.9912

-15.4972482

15.58725

0.02250

1.8061

-7.74862

7.79362

Factor

Effect Estimates; Var.:Enzyme activity (%); R-sqr=,98673; Adj:,96461 (Design: 2**(3-0) design 2**(3-0) design; MS Residual=8,458545
DV: Enzyme activity (%)

Effect

Std.Err.

t(2)

p

-95,% (Cnf.Limt)

+95,% (Cnf.Limt)

Coeff.

Std.Err. (Coeff.)

-95,% (Cnf.Limt)

+95,% (Cnf.Limt)

Mean/Interc.

81.7518

1.1682372

69.978749

0.00020414

76.7253

86.7782968

81.7517778

1.1682372

76.7253

86.7782968

(1)Molar ratio

-19.5163

2.47820534

-7.87515453

0.01574454

-30.1791

-8.85339303

-9.758125

1.23910267

-15.0896

-4.42669652

1 by 2

15.5653

2.47820534

6.28085565

0.02442422

4.9024

26.228107

7.782625

1.23910267

2.4512

13.1140535

(2)pH

14.9113

2.47820534

6.016955

0.02652729

4.2484

25.574107

7.455625

1.23910267

2.1242

12.7870535

(3)Temperature

9.5423

2.47820534

3.85046786

0.06131099

-1.1206

20.205107

4.771125

1.23910267

-0.5603

10.1025535

2 by 3

2.6098

2.47820534

1.05308061

0.4027547

-8.0531

13.272607

1.304875

1.23910267

-4.0266

6.63630348

1 by 3

0.6363

2.47820534

0.25673821

0.82137825

-10.0266

11.299107

0.318125

1.23910267

-5.0133

5.64955348

Table S5. Mixture design-detail statistical results

Factor

Coeffs (recoded comps); Var.:DS; R-sqr=,9929; Adj:,9717 (3 Factor Constrained Mixture (3 Factor Constrained Mixture ([No active dataset]) in Constrained mixtures) in Constrained mixtures)
3 Factor mixture design; Mixture total=100,, 9 Runs
DV: DS; MS Residual=92,81305

Coeff.

Std.Err.

t(2)

p

-95,%

+95,%

(A)A

178.187

9.5949

18.57108

0.002887

136.90

219.470

(B)B

10.383

9.5949

1.08214

0.392307

-30.90

51.666

(C)C

6.220

10.0419

0.61938

0.598823

-36.99

49.427

AB

-280.019

47.1869

-5.93425

0.027242

-483.05

-76.990

AC

-2.723

51.9775

-0.05240

0.962976

-226.36

220.918

BC

40.850

51.9775

0.78592

0.514240

-182.79

264.491

ABC

-447.191

352.0373

-1.27029

0.331762

-1961.89

1067.503

 Factor

Coeffs (recoded comps); Var.:DS; R-sqr=,9929; Adj:,9811 (3 Factor Constrained Mixture (3 Factor Constrained Mixture ([No active dataset]) in Constrained mixtures) in Constrained mixtures)
3 Factor mixture design; Mixture total=100,, 9 Runs
DV: DS; MS Residual=61,9603

Coeff.

Std.Err.

t(3)

p

-95,%

+95,%

(A)A

177.964

7.0267

25.32689

0.000135

155.60

200.326

(B)B

10.353

7.8259

1.32296

0.277649

-14.55

35.259

(C)C

5.883

6.3072

0.93278

0.419740

-14.19

25.955

AB

-279.522

37.7693

-7.40078

0.005103

-399.72

-159.324

BC

41.920

39.0572

1.07329

0.361793

-82.38

166.217

ABC

-455.060

260.1446

-1.74926

0.178559

-1282.96

372.836

 Factor

Coeffs (recoded comps); Var.:Pdi; R-sqr=,9817; Adj:,9269 (3 Factor Constrained Mixture (3 Factor Constrained Mixture ([No active dataset]) in Constrained mixtures) in Constrained mixtures)
3 Factor mixture design; Mixture total=100,, 9 Runs
DV: Pdi; MS Residual=,0007926

Coeff.

Std.Err.

t(2)

p

-95,%

+95,%

(A)A

0.34727

0.028039

12.38524

0.006456

0.22663

0.467907

(B)B

0.09636

0.028039

3.43677

0.075234

-0.02428

0.217004

(C)C

0.34349

0.029345

11.70518

0.007220

0.21723

0.469752

AB

1.03423

0.137893

7.50022

0.017316

0.44092

1.627530

AC

-0.38345

0.151892

-2.52448

0.127568

-1.03699

0.270090

BC

0.12711

0.151892

0.83687

0.490728

-0.52642

0.780654

ABC

-2.40518

1.028748

-2.33797

0.144358

-6.83153

2.021161

 Factor

Coeffs (recoded comps); Var.:Pdi; R-sqr=,9753; Adj:,9342 (3 Factor Constrained Mixture (3 Factor Constrained Mixture ([No active dataset]) in Constrained mixtures) in Constrained mixtures)
3 Factor mixture design; Mixture total=100,, 9 Runs
DV: Pdi; MS Residual=,0007134

Coeff.

Std.Err.

t(3)

p

-95,%

+95,%

(A)A

0.34865

0.026555

13.12904

0.000954

0.26414

0.433159

(B)B

0.10677

0.023843

4.47787

0.020764

0.03089

0.182648

(C)C

0.35920

0.021402

16.78342

0.000461

0.29109

0.427308

AB

1.01106

0.128161

7.88893

0.004245

0.60319

1.418923

AC

-0.43337

0.132531

-3.26991

0.046780

-0.85514

-0.011591

ABC

-2.03789

0.882740

-2.30859

0.104165

-4.84716

0.771387

 Factor

Coeffs (recoded comps); Var.:Zeta; R-sqr=,4942; Adj:0, (3 Factor Constrained Mixture (3 Factor Constrained Mixture ([No active dataset]) in Constrained mixtures) in Constrained mixtures)
3 Factor mixture design; Mixture total=100,, 9 Runs
DV: Zeta; MS Residual=69,48256

Coeff.

Std.Err.

t(2)

p

-95,%

+95,%

(A)A

13.6188

8.3018

1.640465

0.242595

-22.10

49.339

(B)B

6.9348

8.3018

0.835335

0.491423

-28.78

42.654

(C)C

16.0176

8.6886

1.843516

0.206571

-21.37

53.402

AB

-19.4014

40.8277

-0.475202

0.681482

-195.07

156.266

AC

-21.9548

44.9727

-0.488179

0.673699

-215.46

171.547

BC

-41.9080

44.9727

-0.931854

0.449786

-235.41

151.594

ABC

97.5327

304.5945

0.320205

0.779171

-1213.03

1408.097

 Factor

Coeffs (recoded comps); Var.:Zeta; R-sqr=,3962; Adj:,0339 (3 Factor Constrained Mixture (3 Factor Constrained Mixture ([No active dataset]) in Constrained mixtures) in Constrained mixtures)
3 Factor mixture design; Mixture total=100,, 9 Runs
DV: Zeta; MS Residual=33,1751

Coeff.

Std.Err.

t(5)

p

-95,%

+95,%

(A)A

10.8101

4.70099

2.29953

0.069813

-1.274

22.89438

(B)B

5.3952

5.23042

1.03151

0.349592

-8.050

18.84044

(C)C

13.4962

4.60068

2.93354

0.032499

1.670

25.32266

BC

-30.8114

27.10625

-1.13669

0.307192

-100.490

38.86738

  1. “My major concern for this study are the results of the dissolution study. What about different experimental conditions, addition of surfactants? What about in vitrolipolysis and/or digestion study? One of the main reasons for the formulation development was the drug delivery, yet the system has failed to be demonstrated as efficient. Please assess this issue.”

The primary goal of this paper was to provide a proof of concept on the suitability of the solid SEDDSs for lysozyme delivery, especially to demonstrate the usefulness of the QbD concept, to provide some new insights and help quantify complex interactions between material attributes and process variables on critical quality attributes of such formulations, and prove that the integrity and activity of the enzyme may be preserved during the manufacturing process. Therefore, in our dissolution study we focused on the detection the free LYS fraction using “basic” experimental conditions.

We agree with the Reviewer that different experimental conditions in the dissolution study may be advantageous, but the obtained results highlighted a contradiction between the amount of the free lyzozyme fraction and the observed enzymatic activity highlighting that the enzyme partially preserved its activity also in complexed form.

This phenomenon increases the complexity of the dissolution process from such systems, thus motivating us to study it in detail, however the complete understanding of the complex dissociation and its connection to the enzymatic activity are beyond the scope of present article. The method development for the paralell determination of the free and complexed lysozyme amount, and the optimization the whole process would be discussed in details in a follow-up article, where the Reviewer's suggestions on the use of in vitro lipolysis and/or digestion studies as well as comprehensive in vivo studies in animal models to prove its efficiency are planned, along with cytocompatibility studies. We found results of this type of studies suitable for another research paper.

Please find below some new results obtained during the optimization of the dissociation of the HIP complex of lysozyme (LYZ) with Na dodecyl sulphate (SDS).

Method:

  • HIP complexes incubated in four different molar concentrations of NaCl in PBS (0.1, 0.5, 1.0 and 1.5 M) for different time intervals (NaCl can effectively facilitate the dissociation of the complex and increase the portion of the free lysozyme fraction).
  • Recovery (%) = (Amount of the LYZ in the supernatant/Amount of LYZ in the complex) x100
  • The amount of the LYZ in the complex was estimated from the complexation efficiency.
  • The enzymatic activity was determined by the spectrophotometric rate determination (A450, Light path = 1 cm) method using a suspension of lyophilized cells of Micrococcus lysodeikticus
  • The enzymatic activity of the dissociated LYZ is compared with that of the LYZ reference solution having a concentration equivalent to that of 100% dissociated LYZ from the complex.

Results:

Complex prepared at pH 6:

The complexation efficiency = 98.20 % ± 0.55 (average of 50 measurements)

  • After complexation, the complex was dried in an oven (50% fan speed, 25ᵒC temperature) rather than freeze drying (the commonly used method).

Table: Enzymatic activity and the recovered LYZ from the HIP complex prepared at pH 6.

Concentration of NaCl (M)

Enzyme activity (%)24 hr

Enzyme activity (%)48 hr

Enzyme activity (%)72 hr

Enzyme activity (%) 168 hr

Recovery (%) 24hr

Recovery (%) 48 hr

Recovery (%) 72 hr

Recovery (%) 168 hr

0.1

10.22±1.04

9.96±2.41

6.62±1.68

16.02±0.96

1.85±0.23

2.05±0.21

1.87±0.40

1.91±0.09

0.5

14.84±3.87

13.68±1.55

18.44±1.51

18.56±0.55

4.78±0.94

5.76±1.28

4.28±0.59

5.27±0.62

1

17.90±1.55

17.05±0.74

15.95±1.65

24.31±1.29

7.33±2.71

10.63±1.13

9.35±1.92

11.89±2.70

1.5

22.29±4.50

21.75±3.18

24.07±4.11

34.39±4.19

6.80±1.73

16.04±4.12

11.20±2.26

16.56±1.14

Figure-1: Enzymatic activity of the dissociated LYZ from the HIP complex prepared at pH 6.

Figure-2: Recovered LYZ from the HIP complex prepared at pH 6 after incubation in different NaCl molar concentrations solutions.

Observations and comments:

  • As can be seen from the above table and figures, there is a general trend of increasing recovered LYZ as the incubation time increased especially at 1.5 M concentration which gave the highest values.
  • In general, the enzymatic activity followed the same trend in the recovery (%). However, in most time points, again 1.5 M solution showed the highest values, especially after incubation for 168 hr (7 days).
  • Increasing of NaCl concentration led to higher recovery and enzymatic activity.
  • There is no clear correlation between the recovery and enzymatic activity and it appears that the enzymatic activity is much higher than the recovery. This could be explained by that part of the complexed LYZ is enzymatically active and more investigations are required to support this assumption.

Round 2

Reviewer 2 Report

Authors have addressed appropriately all the raised comments.